# Sparse haplotype-based fine-scale local ancestry inference at scale reveals recent selection on immune responses

Yaoling Yang [1,2] ✉, Richard Durbin [3], Astrid K. N. Iversen [4] &
Daniel J. Lawson [1,2] ✉

Increasingly efficient methods for inferring the ancestral origin of genome regions are needed to gain insights into genetic function and history as biobanks grow in scale. Here we describe two near-linear time algorithms to learn ancestry harnessing the strengths of a Positional Burrows-Wheeler Transform. SparsePainter is a faster, sparse replacement of previous model-based 'chromosome painting' algorithms to identify recently shared haplotypes, whilst PBWTpaint uses further approximations to obtain lightning-fast estimation optimized for genome-wide relatedness estimation. The computational efficiency gains of these tools for fine-scale local ancestry inference offer the possibility to analyse large-scale genomic datasets using different approaches. Application to the UK Biobank shows that haplotypes better represent ancestries than principal components, whilst linkage-disequilibrium of ancestry identifies signals of recent changes to population-specific selection for many genomic regions associated with immune responses, suggesting avenues for understanding the pathogen-immune system interplay on a historical timescale.

Modern human populations are complex mixtures between ancient contributing source groups[1]. Genetic admixture is the process of mixing groups that were genetically distinct due to genetic drift, which can create new distinct populations[2,3]. The process is ubiquitous and spans scale in space and time, from the admixture with Neanderthals around 50,000 years ago when modern humans migrated out of Africa[4], to native Americans mixing with primarily European and African immigrants over the last 500 years to form the majority of United States ancestry[5], and the fine-scale geographical regionalisation within a single country such as the UK[6]. The identification of chromosomal regions originating from a specific population is known as local ancestry inference (LAI)[7], which can be used to map disease loci[8], investigate the relationships between modern populations, improve association studies[9], and study demographic histories[10].

Genome-wide association studies (GWAS) have identified single nucleotide polymorphisms (SNPs) associated with human complex traits and diseases[11], but the SNP frequencies are likely to be associated with particular ancestries. Local ancestry may then either be viewed as a confounder of the SNP effect[9], or treated as a predictor as in 'Ancestral GWAS'[12]. In this framing, local ancestry inference examines the ancestral origin of risk loci in terms of a population and a time – for instance, risk alleles associated with multiple sclerosis originated from pastoralists dwelling on the Pontic Steppe, which were brought into Europe by the Yamnaya-related migration around 5,000 years ago[12]. Other examples include the relationship between platelet count in Hispanics and an Amerindian-origin variant of the ACTN1 gene[13], a link between quantitative red blood cell traits and African- and Amerindian-origin loci in the HBA1/2 gene[14], and kidney disease in African-origin variants of the APOL1 gene[15].

[1]Department of Statistical Sciences, School of Mathematics, University of Bristol, Bristol, UK. [2]MRC Integrative Epidemiology Unit, Population Health Sciences, University of Bristol, Bristol, UK. [3]Department of Genetics, University of Cambridge, Cambridge, UK. [4]Nuffield Department of Clinical Neurosciences, John Radcliffe Hospital, University of Oxford, Oxford, UK. ✉e-mail: yaoling.yang@bristol.ac.uk; dan.lawson@bristol.ac.uk

It is hard to perform LAI accurately and efficiently. Various LAI software have been developed since the 21st century, and the majority[16] are based on the Li and Stephens hidden Markov model (HMM)[17], including HAPMIX[7], ChromoPainter[18], LAMP-LD[19], MOSAIC[3] and FLARE[20]. HAPMIX pioneered this application but is limited to modelling two ancestries. In comparison, ChromoPainter enables the accurate analysis of admixtures from multiple groups but is slow. LAMP-LD is faster but can be unstable[16]. The distinctive feature of MOSAIC is that the knowledge of the intricate connections between reference haplotypes and ancestral mixing groups is not required[3]. Recently, through the on-the-fly compression of reference panels, saved checkpoints and composite reference haplotypes, FLARE greatly improves the computational performance compared with the previous LAI software[20]. Other approaches for local ancestry inference are also possible, among which PCAdmix, a Principal Components-based algorithm[21], and RFMix[22], which employs a discriminative modelling strategy, are popularly used.

Our technical contribution is providing two scaleable algorithms for identifying fine-scale population structure under different use cases. This allows application to hundreds of thousands or even millions of samples as presented by the most challenging modern biobanks and association studies. These approaches avoid storing the entire genotype information in memory, instead using the Positional Burrows-Wheeler Transform (PBWT)[23,24] to extract only a sparse set of the longest haplotype matches to the reference panel at each position. In PBWTpaint[25], only the longest set-maximal matches are retained, which we will show is sufficient for estimating genome-wide ancestry. In SparsePainter[26], we extract a richer set of haplotypes on which we show that a sparse implementation of the Li and Stephens HMM model[17] can be run with a negligible accuracy cost by using a Hash Map data structure[27].

Identifying genomic features that are of biological significance from fine-scale local ancestry information is an under-explored topic and the core of our scientific contribution. Within SparsePainter we are able to efficiently compute Linkage Disequilibrium of Ancestry (LDA), LDA score (LDAS) and Ancestry Anomaly Score (AAS)[12] at scale. These recently proposed summary statistics of local ancestry are predicted under recent population-specific selection, but previous implementations based on post-processing local ancestry data are only suitable for examining small sections of genome or small reference datasets. LDA is the correlation of ancestries between SNP pairs, which measures whether recombination events between ancestries are more frequent than those within ancestries. LDAS calculates the total LDA of each SNP on the chromosome weighted by genetic distance. A lower LDAS indicates the haplotype inherited from the reference population is shorter than expected. We identify two mechanisms that generate low LDAS and both involve a change in selection between the pre-existing and admixed population. The first involves selection on a nearby locus, leading to balancing selection at the level of haplotypes. The second is against a locus that was high frequency in at least one contributing population. AAS is the degree of difference between the estimated average ancestry probabilities and the genome-wide average, which detects signals of recent selection for loci experiencing changes in ancestry frequencies.

We benchmarked SparsePainter against FLARE, ChromoPainter, RFMix, and MOSAIC, which demonstrates that SparsePainter is faster both empirically and in scaling at fine scale, i.e. as the number of reference populations grows. PBWTpaint is faster than all methods (including a recent innovation Neural ADMIXTURE[28]) by orders of magnitude in identifying genome-wide haplotype structure within a single dataset, which is its specific capability.

In exploring population structure within the UK Biobank (UKB) with PBWTpaint, we construct haplotype principal components (HCs) which we compare to the widely-used SNP-based principal components (PCs). HCs are better associated with birthplace and seem to capture more nuanced genetic variation than PCs, revealing distinct ancestral patterns among ethnic backgrounds and significant regional distinctions within the UK and Ireland, suggesting potential for more refined population stratification in genetic studies. Using 1000 Genomes Project (1000GP) Data[29] as reference, we can apply the LDAS and AAS statistics to identify genes that show signals of recent changes to population-specific selection. This approach, applied genome-wide, identifies a number of genes that are almost entirely immune-related, pointing to population-specific immune responses as a central driver of selection acting on historical timescales.

## Results
### Method overview
There are two main approaches to ancestry inference. The first is unsupervised learning, which addresses the goal of learning fine-scale population structure. Examples include clustering[18], unsupervised admixture models[1,30], or dimensionality reduction such as Principal Component Analysis (PCA) based on either genotype[31,32] or haplotype data[18]. Here, the data are not typically curated and we aim to form the largest dataset possible for the analysis. The second approach is supervised learning, in which target individuals are compared to carefully curated reference populations, and recently admixed individuals (which are the majority of individuals) are not directly used. The goal of supervised learning divides into ancestry estimation which can be used analogously to unsupervised genome-wide ancestry profiles[33], or local ancestry estimation in which the ancestry of particular sections of DNA is inferred.

These goals are met by two tools that facilitate haplotype-based ancestry analysis at scale, as described in Fig. 1. The first of these is PBWTpaint, a direct extension of the PBWT[23] which rapidly identifies long matches. PBWTpaint only considers a very limited subset of possible matches representing the maximally shared haplotypes at any locus (called set-maximal). Further, the Li & Stephens model is replaced with an approximation that only considers overlapping set-maximal matches, running in linear-time so that large-scale analyses are straightforward. Larger datasets uncover longer, more recent matches, and any inaccuracies due to modelling approximations average out over the whole genome for genome-wide analyses.

The second tool is SparsePainter, which is designed to perform accurate local ancestry inference efficiently. Whilst SparsePainter can perform all-vs-all painting, it is optimised for either painting a reference panel against itself (reference-vs-reference painting), or painting target individuals using a reference panel (target-vs-reference painting). There are two primary outputs of SparsePainter. The first is local ancestry estimates, which are the probabilities that a haplotype at a particular chromosomal location is inherited from each ancestral individual or population. By efficient representation of this, we can efficiently compute the selection statistics LDA, LDAS and AAS. The second is the expected fraction of the total genome shared most recently between a target and each reference ancestral individual or population, as used in complex admixture history modelling, e.g. the GLOBETROTTER tool[1].

Chromosome Painting is conceptually different to identifying haplotypes identical-by-descent (IBD); it assigns every position of the genome to the most recent common ancestor in the reference, without allowing overlaps or conditioning on length and hence the expected age of a sharing event. This facilitates fine-scale ancestry estimation or 'Admixture Modelling'[1,18] using the expected fraction of the total genome shared most recently between a target and each reference ancestral individual or population. As we use a leave-one-out scheme to make individuals from the reference and target datasets exchangeable, i.e. receive the same ancestry inference if they share the same ancestry (see Methods for a formal definition), this allows population history reconstruction without assuming perfect references[1].

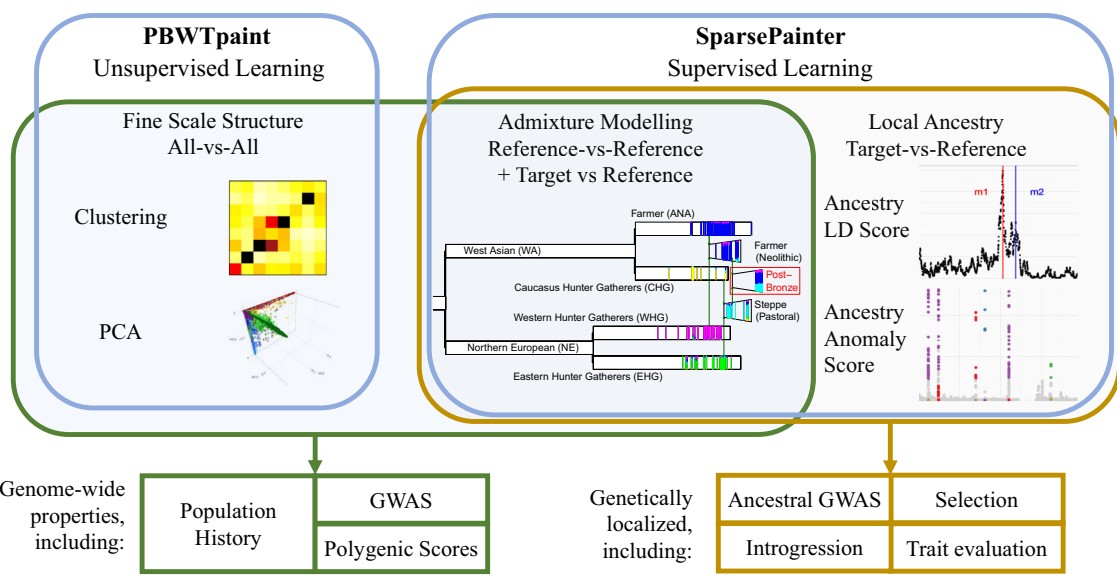

**Fig. 1 | An overview of the scientific use cases of SparsePainter and PBWTpaint.** PBWTpaint performs unsupervised all-vs-all painting, for use in fine-scale structure estimation such as clustering and PCA. SparsePainter performs supervised learning which allows reference-vs-reference painting for admixture estimation and population history modelling (plots redrawn from Barrie et al.[12]). It also allows target-vs-reference painting for local ancestry inference, with uses including selection detection via LDAS and AAS.

## PBWTpaint

Storing the genotype information of all the samples in memory is a problem for large datasets. The PBWT[23] is a data structure to transform a binary matrix $X_{ik}$ (with $2N$ haplotypes and $K$ SNPs) into a sequence of run-length compressed arrays per SNP, in each of which the haplotype values at the SNP are sorted according to the reversed haplotype prefixes preceding the SNP. From a PBWT, long matches can be efficiently extracted using the ReportMatches algorithm, and set-maximal matches with the ReportSetMaximalMatches algorithm, in $O(NK)$ operations for all haplotypes at the same time. Our models are built on these matches.

For each target individual $i$, PBWTpaint iterates through the $M(k)$ matches at a locus $k$ (which are typically very sparse, and sparse by construction for set-maximal matches). For each matched reference haplotype $j$ we extract the start $s_{jk}$ and end $e_{jk}$ positions of the maximal exact match to $j$ covering $k$, i.e. $s_{jk}$ is the location just after the first upstream mismatch, and $e_{jk}$ is the location just before the first downstream mismatch. From these, we compute a weight $w_{jk} = (k - s_{jk})(e_{jk} - k)$, i.e. the weight increases linearly with distance from each end of the match, and quadratically with the total length of the match for positions at the midpoint of the match. This is normalised over matches $j$ to give a local ancestry score $p_{jk} = w_{jk} / \sum_{l=1}^{2N} w_{lk}$, which we sum over loci $k$ to produce a genome-wide ancestry estimate $p_j$. We also provide estimates of the total number of recombination events, as well as regional bootstraps, to enable clustering with FineSTRUCTURE[18].

### From PBWT to accurate sparse local matches

For local ancestry inference, the longest haplotype matches at the target locus are the most important, since short matches appear within any ancestry due to statistical noise and incomplete lineage sorting, i.e. ancient structure shared across ancestries rather than recent genealogical relationships. As such, it is essential to find the longest matches at a given locus, even if they are short relative to other places in the genome.

Whilst the original PBWT algorithm finds long matches only within the same database, it has been extended to report long matches between different haplotype sets[24]. For accurate and efficient local ancestry inference we detect all matches longer than some threshold $L$, but there may be no genome-wide 'correct' $L$. Some target haplotypes will only have short matches if they diverged a long time ago, and few or even no matches are longer than $L$. Other target haplotypes will share very long segments of DNA with many reference haplotypes leading to many matches being retained, the shorter of which (also longer than $L$) are not helpful for inferring ancestry. To ensure enough matches are found even in genome regions without long matches, we extend the 'ReportLongMatches' algorithm of PBWT with a 'ReportQLongestMatches' algorithm which aims to find at least $Q$ longest matches at each position for a target sample $i$ (Methods). With this algorithm, we maintain a particular sparsity level at each location while also preserving the longest matches to guarantee accuracy.

### Using Hash Map to perform HMM forward-backward algorithm in sparse form

SparsePainter stores haplotype matches in a Hash Map data structure that implements an associative array abstract data type for efficient key-value storage and retrieval[27], facilitating $O(1)$ storage and lookup of values (here painting probabilities) based on unique identifiers or keys (here haplotype indices). We then employ a sparse approximation to Li and Stephens[17] model by vectorising the forward and backward probabilities and assuming a vanishing mutation rate (Methods). The forward and backward computation is only required within the $Q$ longest matches to the target haplotype at each locus, allowing efficient computation of the local ancestry probabilities and the expected genome shared. Compared with computing and storing the probabilities at all $N$ haplotypes, our approach reduces both memory usage and compute time from $O(N)$ to $O(Q)$.

### Simulation overview

We used SLiM (v3.7.1)[34] to simulate genetic data on 20 megabases throughout 3000 generations, aiming to compare the accuracy, speed and memory utilisation of SparsePainter (v1.2.1), ChromoPainter (implemented in FineSTRUCTURE (v4)), FLARE (v0.5.1), PBWTpaint (v3.0-e24809b), RFMix (v2.03-r0), MOSAIC (v1.5.1), and Neural ADMIXTURE (v1.4.1) in terms of local ancestry and (or) genome-wide estimates. Because there are multiple distinct uses of our software we have designed, four distinct simulation models with 20k SNPs (noting that all methods have linear compute and memory requirements in the genome size analysed; see Methods for details):

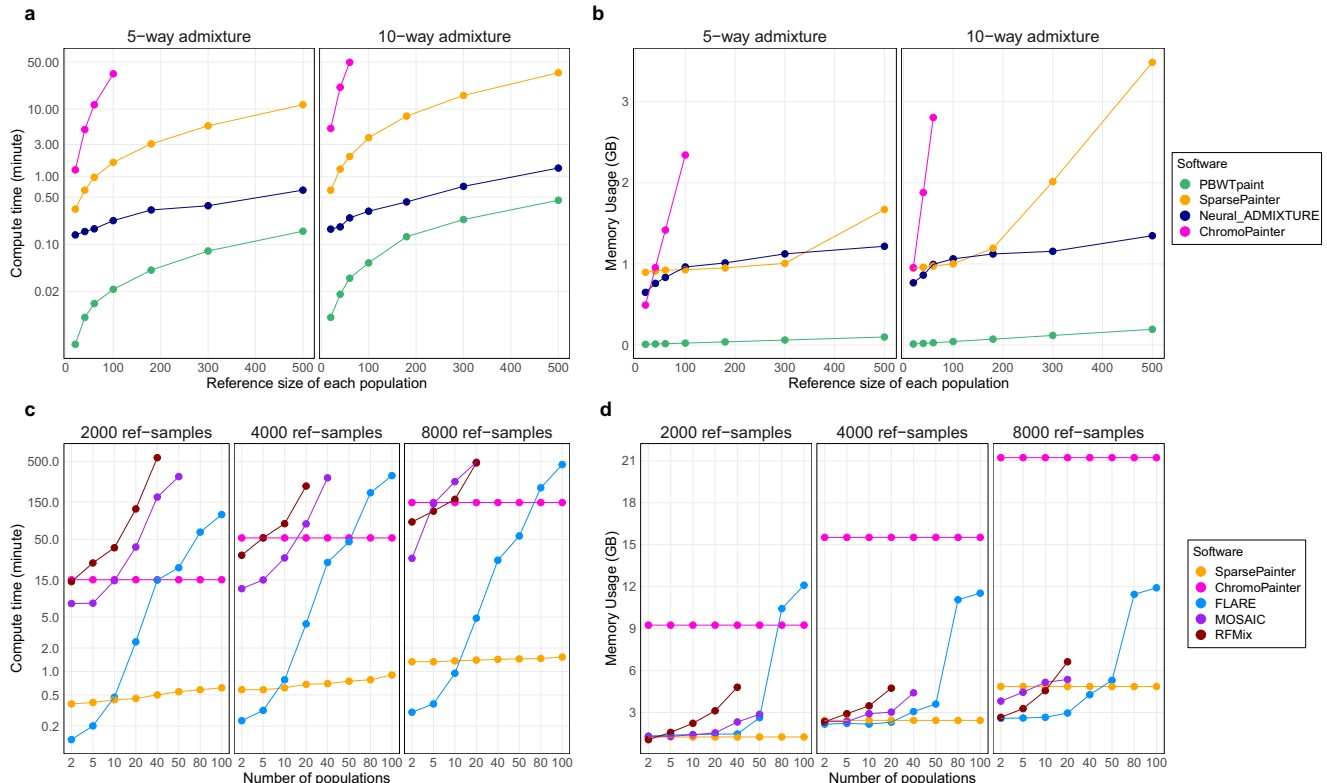

**Fig. 2 | Speed and memory comparison between software. a**, **b** under Simulation 1: comparing software that can perform reference-vs-reference analysis with 5 or 10 populations and different reference sizes with 20k SNPs. **c**, **d** under Simulation 2a: comparing software that can compare 50 target individuals to a reference with different numbers of populations and reference sizes with 20k SNPs.

(1) Simulation 1: A hierarchical model designed to assess the speed, memory usage, and accuracy of PBWTpaint, SparsePainter, ChromoPainter, and Neural ADMIXTURE for reference-vs-reference (supervised) and all-vs-all (unsupervised) analysis;

(2) Simulation 2a: An evolutionary process designed to investigate the scaling of SparsePainter, FLARE, ChromoPainter, RFMix, and MOSAIC in target-vs-reference painting by generating 2 to 100 different populations;

(3) Simulation 2b: As Simulation 2a but with smaller and fewer populations, designed to assess the accuracy of target-vs-reference painting for SparsePainter, FLARE, ChromoPainter, RFMix, and MOSAIC;

(4) Simulation 2c: As Simulation 2b but with larger populations and more target individuals, designed to investigate how Sparse-Painter balances accuracy against speed and memory utilisation in target-vs-reference painting.

## Within-sample performance comparison

We first compared the efficiency of PBWTpaint (using all-vs-all painting), SparsePainter, ChromoPainter, and Neural ADMIXTURE (using reference-vs-reference painting) under Simulation 1. FLARE is excluded as it can neither perform within-sample (i.e. reference-vs-reference), nor genome-wide, comparisons. Performance is measured using the recovery rate of an individual's own population ancestry fraction using squared Pearson's correlation coefficient (denoted as $r^2$) with the truth (Methods). Neural ADMIXTURE is excluded for accuracy comparison because it does not estimate local ancestry.

Figure 2a, b show the computational scaling of each method. ChromoPainter in theory has a quadratic cost as a function of panel size, and scales poorly to larger reference sizes. SparsePainter is close to linear in both speed and memory efficiency regardless of reference

sizes. Neural ADMIXTURE has an almost linear compute time with constant scaling in memory, which is more efficient than Sparse-Painter. Whilst PBWTpaint also scales linearly, it is orders of magnitude faster and requires significantly less memory than Neural ADMIXTURE, and only introduces a minor trade-off in terms of accuracy (Fig. 3a). Notably, PBWTpaint only retains accuracy for genome-wide estimation, as its simple model with set-maximal matches isn't suitable for estimating local ancestries (Methods).

## Target-vs-reference speed and memory comparison for LAI

As PBWTpaint can neither paint target samples against different reference panels, nor perform local ancestry estimates, we restricted our speed and memory comparison to SparsePainter, ChromoPainter, FLARE, RFMix, and MOSAIC with Simulation 2a. As all those software are based on the Li and Stephens hidden Markov model, their computational costs for genome-wide and local ancestry estimates are expected to be similar.

The speed and memory of SparsePainter and ChromoPainter remain largely unaffected by the number of true populations, because both of them perform chromosome painting that is ancestry-unaware. Conversely, whilst FLARE demonstrates impressive speed and efficient memory usage with few populations ($n_{pop} \le 5$), its efficiency dramatically diminishes compared to SparsePainter when handling 20 or more populations (Fig. 2c, d). When painting with 100 populations, Sparse-Painter is over 100 times faster and 10 times more memory-efficient than FLARE. As in previous work[20], we conclude that RFMix and MOSAIC are slow compared with other tools, especially at fine scale.

A recent study[33] decomposed the UK Biobank into over 100 distinct fine-scale reference ancestries. We replicated their analysis with 4334 reference individuals from non-restricted data sources (i.e. all except POPRES) spanning 129 populations. For 1000 target individuals

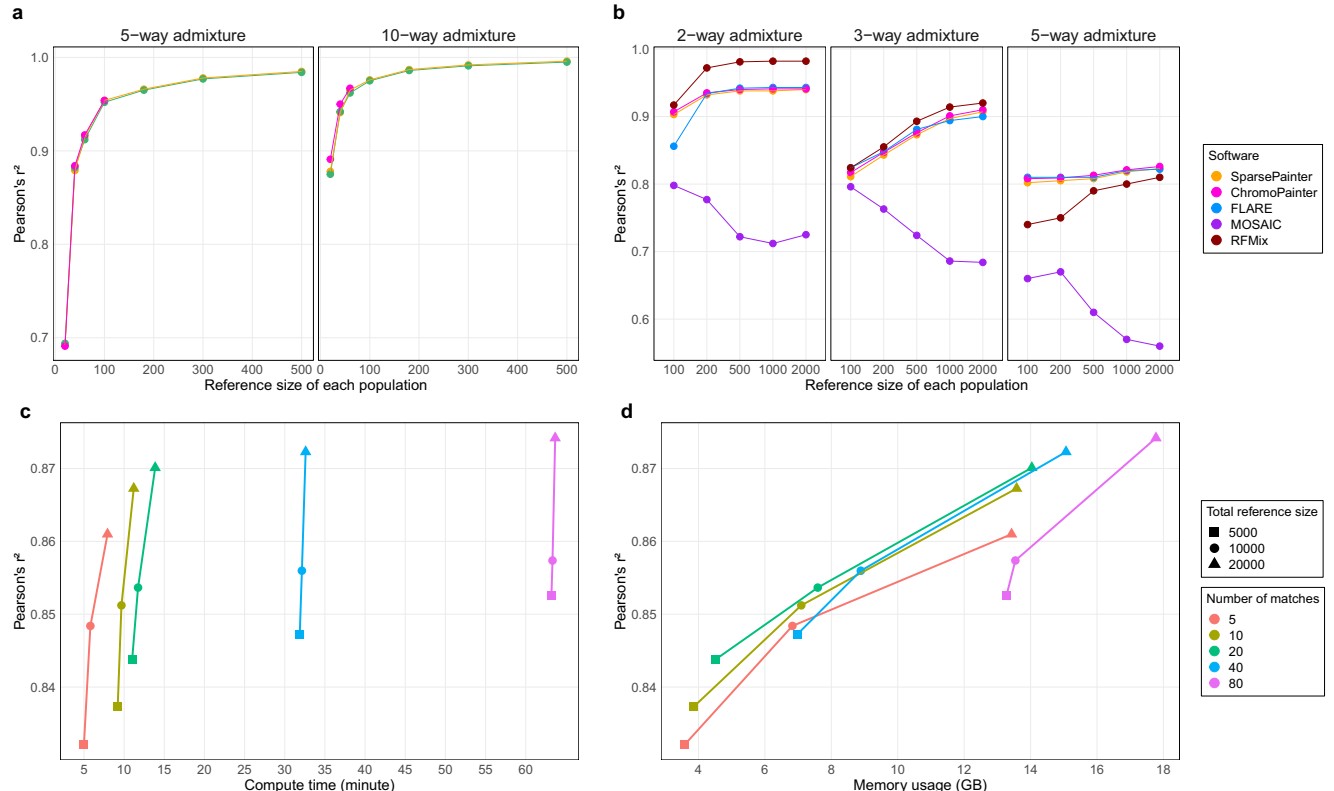

**Fig. 3 | Accuracy in simulated data of local ancestry recovery as measured by the correlation with the truth. a** Under Simulation 1: reference-vs-reference analysis using 20k SNPs. **b** Under Simulation 2b: target-vs-reference analysis using 20k SNPs and 50 target individuals sampled 13 generations after admixture. **c**, **d** Under Simulation 2c: compute time and memory usage of SparsePainter for target-vs-reference analysis of 1000 target individuals with different sparsity and reference sizes under a 5-way admixture model with 20k SNPs.

on chromosome 21 which comprises 9522 SNPs, SparsePainter is dramatically faster and requires minimal memory (6 minutes and 1.5GB) compared with ChromoPainter (272 minutes and 10.2GB) and FLARE (338 minutes and 14.2GB).

### Target-vs-reference accuracy comparison for LAI and admixture estimation

We have illustrated the circumstances when SparsePainter has superior speed and memory use than FLARE, ChromoPainter, RFMix, and MOSAIC, but it is crucial to maintain accuracy. In Simulation 2b we examined the accuracy of local ancestry estimated by both the squared Pearson's correlation coefficient and the proportion of accurate local ancestry predictions (Methods). Across all software except MOSAIC, the accuracy of local ancestry estimation consistently improves with increased reference sizes and reduced number of populations (Fig. 3b and Supplementary Fig. 1).

Whilst the accuracy of SparsePainter and FLARE is comparable, we note that FLARE restricts reporting of LAI to SNPs that it can find IBD. The excluded SNPs have less certain LAI due to short matches, which can be caused by genotyping error or ancient admixture, so its reported accuracy is an upper bound. Also as anticipated, Sparse-Painter displays a negligible accuracy drop compared to Chromo-Painter, given that SparsePainter is essentially a sparse implementation of ChromoPainter. RFMix has higher accuracy than SparsePainter only when $n_{pop} \leq 3$, and MOSAIC has the lowest accuracy because it uses an unsupervised algorithm that reconstructs mixing populations which is not appropriate for supervised simple two-way admixture. We have also shown comparable accuracy for admixture estimation between SparsePainter and ChromoPainter (Supplementary Note 2 and Supplementary Fig. 2).

We show using simulation (Methods) that SparsePainter can tolerate genotyping error (Supplementary Table 2) up to around 0.1–0.2%. This is superior to FLARE, and SparsePainter only has negligible accuracy loss compared with ChromoPainter under small reference sizes (Supplementary Table 3), different phasing errors (Supplementary Table 4), high recombination rates (Supplementary Table 5), and different admixture times for target individuals (Supplementary Table 6). By showing the local ancestry signal is still preserved to distant admixture times (Supplementary Table 6), e.g. the proportion of accurate LAI retains 93.9% at 150 generations, we can explain why modelling of complex ancient admixture remains robust[1].

### Sparsity in SparsePainter

To investigate SparsePainter's tradeoff between sparsity and accuracy, we varied the reference size of a 5-way admixture model (Simulation 2c). Figure 3c, d shows that a larger reference size substantially boosts accuracy, whilst increments in the number of matches only marginally elevate it. Using $Q = 20$ matches saturates accuracy whilst minimising speed and memory cost, and larger reference samples dilute the accuracy's sensitivity to sparsity. However, SNP density may influence the exact $Q$ value needed for optimal performance. Conversely, computational time and memory demands surge considerably as match density escalates. This indicates that if large reference datasets are available, opting for a constant number of matches (so diminished match proportion) yields significant computational savings, at a negligible compromise in accuracy.

### Haplotype principal components analysis for the UK Biobank

The UK Biobank's PCs are widely used for correctly inferring the population structure. We inferred the (sparse) genome-wide pairwise

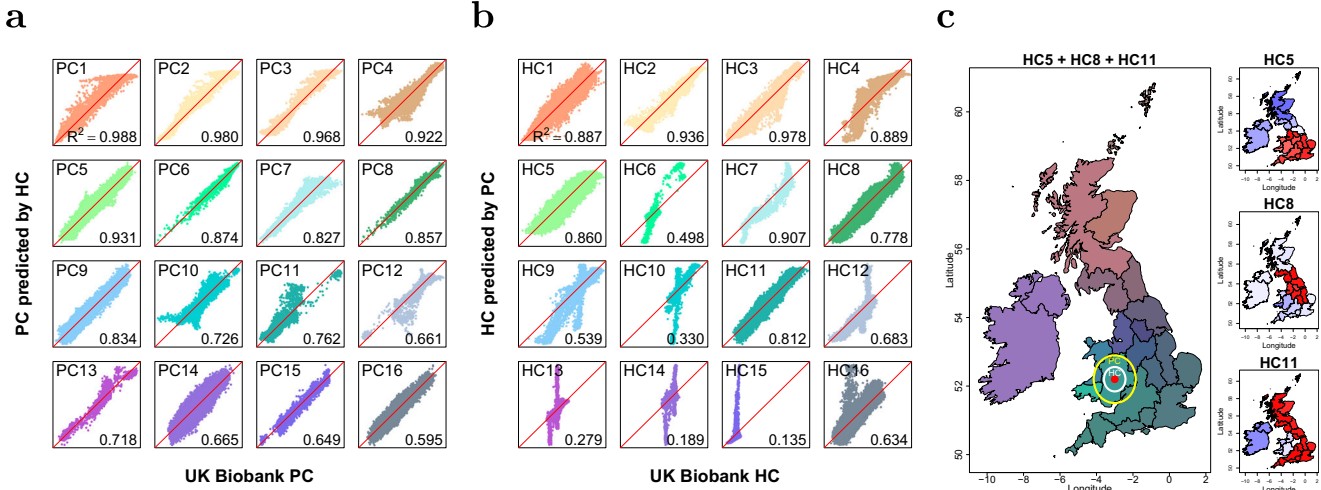

**Fig. 4 | Comparison between UK Biobank PCs and HCs and the decomposition of HCs. a** The coefficient of determination for predicting the first 16 UKB PCs (*y*-axis) using the first 150 HCs (*x*-axis) with linear regression models (*n* = 406,733 individuals), which shows strong correlations. **b** the coefficient of determination for predicting the first 16 UKB HCs (*y*-axis) using the first 150 PCs (*x*-axis) with linear regression models (*n* = 406,733 individuals), which shows strong correlations on only 12 of the first 16 HCs. **c** Visualisation of the average of the 5th, 8th and 11th HC stratified by birthplaces within the UK and Ireland (*n* = 347,532 individuals), corresponding to the red, green, and blue channels, respectively, in the composite plot (left), and the right plot shows the decomposition of each HC. We have also shown the median prediction error range of the birthplace of HCs (white circle, radius 39.7 km) and PCs (yellow circle, radius 77.5 km) of an east Wales location (red point).

coancestry of *N* = 406, 733 UK Biobank individuals via PBWTpaint from *L* = 569, 200 SNPs, taking 41 CPU hours (which is parallizable and scales as $O(NL)$). We summarised these ancestries into the top 150 HCs (Methods), and compared their informativeness with PCs in several ways. First, we can accurately predict the first 16 PCs with the first 150 HCs using linear regression models (Fig. 4a), especially for the first 9 PCs which have a coefficient of determination ($R^2$) exceeding 80%. Conversely, when using the first 150 PCs to predict the first 16 HCs, some of the HCs are poorly explained (Fig. 4b). This observation indicates that HCs might encapsulate additional ancestry information beyond that conveyed by PCs.

To investigate consistency across chromosomes, we split the SNPs from the odd and even chromosomes and then computed the top 150 PCs and HCs from the even chromosome set. Subsequently, we used 150 HCs/PCs from one set to predict each of the top 50 HCs/PCs from the other set. HCs are well explained with $R^2 > 90\%$ for the majority of them (Supplementary Fig. 3), indicating HCs capture ancestry information that is shared in all the chromosomes. By contrast, few PCs can be predicted from different chromosome sets, which corresponds to the previous finding that all PCs except the top few of them are related to specific genetic regions[35].

HCs are associated with self-reported ethnicity (Supplementary Fig. 4): the 2nd and 3rd HCs effectively differentiate within white and black backgrounds, respectively, whereas the 4th and 6th HCs reflect variations associated with South Asian ancestry. HCs are also associated with geography: filtering to the 347,532 individuals with white, British or Irish ethnicity born in the UK or Ireland, we plotted the average HC for 23 UK regions (Supplementary Fig. 5). The 5th HC represents the variation between Scotland, Irish, and the rest of the UK, while the 11th HC differentiates Ireland and Wales from the other regions. By mapping the 5th, 8th, and 11th HCs onto the geography of the UK and Ireland, we created a colour-coded depiction (Fig. 4c) which uniquely identifies each county. We demonstrate through simulations that only 5 HCs can almost perfectly predict genome-wide ancestries, outperforming the predictive power of even 150 PCs (see Methods and Supplementary Table 7).

Further, predicting birth location using HCs has a median error of 39.7km, whilst PCs give a nearly double error of 77.5km in out-of-sample individuals (see Methods). This is a surprisingly high accuracy

as these individuals were not filtered for having ancestry from a single location, so prediction accuracy is bounded by migration since people need not be born where their ancestors came from.

## Ethnicity-specific selection in the UK Biobank compared to the 1000 Genomes populations

To demonstrate the scientific value of SparsePainter, we inferred the local ancestry of 487,409 UK Biobank[36] individuals using the 2504 individuals spanning 26 populations from the 1000 Genomes Project[29] as reference. From this, we evaluated selection using LDA score, which quantifies genomic regions with particularly short ancestry segments, compared to the base recombination rate, as well as AAS, which quantifies regions of unusual ancestry, compared to genome-wide (see Methods). We report results that replicate over 7 primary self-reported ethnic backgrounds (hereafter ethnicities) within the UK Biobank: British, Irish, Indian, Caribbean, African, Pakistani, and Chinese. The LDAS, AAS and average probabilities of 26 1000GP populations for each SNP analysed within each ethnicity are available in Supplementary Data 1–7.

Our goal is to demonstrate applications of local ancestry at scale outside of population history and admixture estimation (Fig. 5a, Methods). We look for signals of 'putative selection' in the form of low LDAS and unusual AAS that are shared, i.e. identified in every UKB primary ethnicity, after extensive quality control (Methods). As a sensitivity analysis, we further painted the UK Biobank with 1000GP data using 5 continental ancestries (EUR, AFR, SAS, EAS and AMR). The LDAS and AAS results of different UKB ethnicities are illustrated in Fig. 5b, c. These are mapped to genes, with shared significant low LDAS and AAS signals visualised in Fig. 6 and investigated in detail in Supplementary Note 1. Genes with ethnicity-specific AAS signals are reported in Supplementary Table 1.

To aid in interpreting these signals, we extended simulations for low LDAS from Barrie et al.[12] (Methods, Supplementary Figs. 6, 7). Two scenarios produce significantly low LDAS and high AAS, and both imply a change in selection following admixture. One scenario is single-locus negative selection in the admixed population, following non-negative selection in the pre-existing populations. The second scenario is multi-locus positive selection in the admixed population, while those loci are either absent or present at low frequency in some of the pre-existing populations. Selection under these scenarios is not

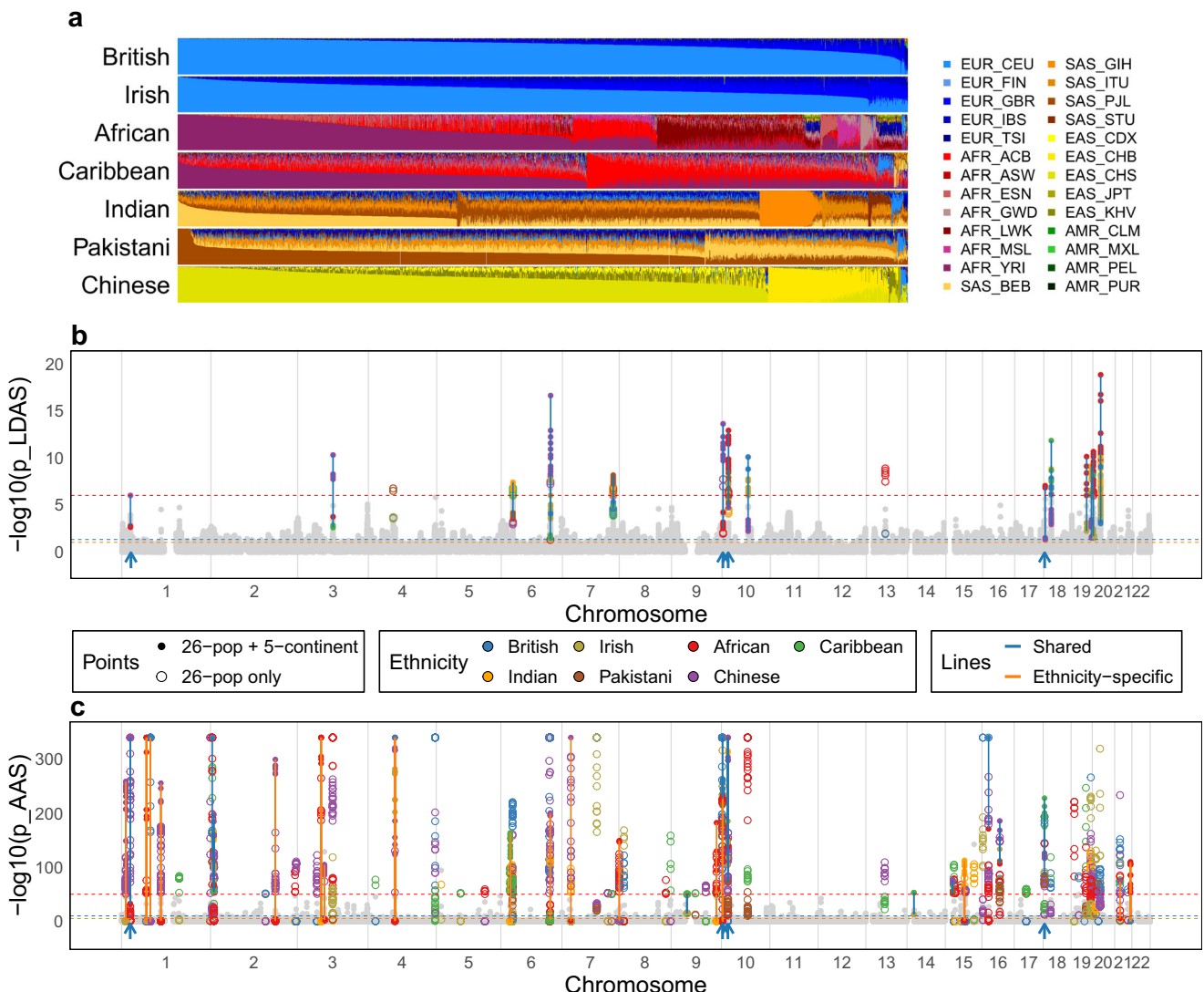

**Fig. 5 | Modelling of 7 UK Biobank self-reported ethnicities using 26 1000GP populations. a** Overall ancestry inference stratified by UKB ethnicities. For each ethnicity, the column shows ancestry decomposition for a single individual, with colours representing different 1000GP reference populations, named as regions followed by local population in standard abbreviation[29]. **b** Linkage-Disequilibrium of Ancestry Score (LDAS), reporting −log10 of the *p*-value of low LDAS (one-sided normality test). **c** Ancestry Anomaly Score (AAS) as a function of genome position, reporting −log10 of the *p*-value of AAS (two-sided chi-squared test, −log10(*p*) is capped at 340 in the plot). All plots describe the analysis of *n* = 487,409 individuals on 569,200 SNPs. In (**b**, **c**), the non-light-grey points (light grey points) represent the SNPs' maximum and minimum values that exhibit significant (insignificant) scores in both (either) paintings with 1000GP 26 populations and 5 continents, respectively (Methods), and blue (orange) lines connect the maximum and minimum values at each SNP that are shared (ethnicity-specific) across the 7 ethnicities in both paintings. The thresholds used to determine significance are depicted as horizontal lines in dashed red, blue and orange, respectively.

detected by iHS, iHH12 or nSL as calculated using selscan[37], showing that significant LDAS SNPs are not expected to be previously reported as under selection.

The most significant AAS signals in all 7 UKB ethnicities (Supplementary Fig. 8) include LINC01432 (Long Intergenic Non-Protein Coding RNA 1432) from chromosome 20 (linked to retroperitoneum carcinoma and early-onset androgenetic alopecia) which has an exceptionally high Japanese ancestry (EAS_JPT) across all UKB ethnicities. Similarly, LINC03004 (highly expressed in testis and the gall bladder) and its nearby gene PTPN11P3 on chromosome 6 are predominantly represented by African ancestry across all ethnicities, a striking example of which is seen in Chinese ethnicity, where LINC03004 is almost completely African. Likewise, in the genes PNRC2 and SRSF10 on chromosome 1, the Puerto Rican ancestry (AMR_PUR) is over-represented, particularly within European and Asian ethnicities.

We observed that the different selection patterns of genes associated with the immune system were related to distinct hierarchies of control of immune response, from control of gene expression to T cell receptor recognition and inflammation. At the core were genes with low LDAS and AAS signals in both the 26 population ancestries and the 5 continental ancestries. These genes affect RNS degradation (PNRC2) and RNA splicing (SRSF10), and include a receptor that binds high-mannose structures on the surface of potentially pathogenic viruses, bacteria, and fungi (MRC1). The product of these genes affects immune responses (Supplementary Note 1.1), but their function is also central to non-immune pathways, and mutations in these genes can give rise to, for example, various congenital disorders and neurological and metabolic diseases.

The second level of control is broad-impact immune genes with low LDAS and AAS signals only in the (more recently separated) 26 population ancestries. The product of these genes affects antigen

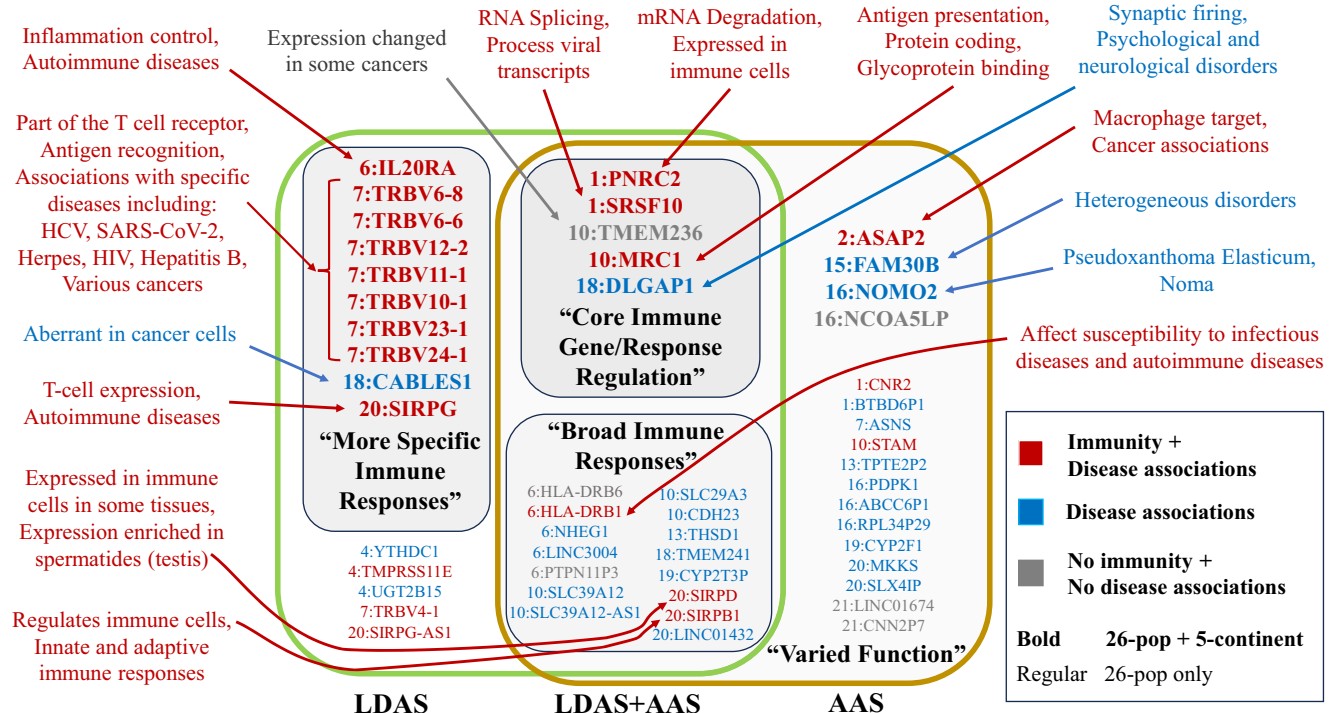

**Fig. 6 | Summary of previous findings for genes with low LDAS and AAS signals shared between 7 UK Biobank self-reported ethnic backgrounds.** Genes with low LDAS and AAS signals in both 26-pop and 5-continent paintings include those with core immune gene or response regulation, while those in 26-pop painting only include many broad-impact immune genes. Genes with LDAS-only signals in both 26-pop and 5-continent paintings more typically affect responses to specific infections, and genes with AAS-only signals have varied functions and disease associations. Classification (colour) and category summaries (bold quoted text) are based on heuristic features of previous work; see Supplementary Note 1 for details.

presentation and the strength of receptor signalling. One of the genes (HLA-DRB1) presents antigens to T cells and helps regulate immune responses. Over 2000 variants of DRB1 have been identified[38], some of which are associated with certain diseases or conditions (autoimmune diseases and susceptibility or protection infection). Whilst HLA-DRB6 is a pseudogene with, as of now, no known function, SIRPB1 encodes a signal-regulatory-protein that interacts with TYROBP/DAP12, a transmembrane adaptor protein on natural killer (NK) cells, peripheral blood monocytes, macrophages, dendritic cells, osteoclasts, and microglia. Through this interaction, SIRPB1 is involved in regulating both adaptive and innate immune responses and other pathways.

The least-central control level primarily affects responses to specific infections (T cell recognition, signalling) or localised responses that occur at the site of infection (inflammation), and have low LDAS scores but no AAS signals. Among them are eight less-commonly expressed TRBV genes, which are noteworthy for well-established associations with globally widespread and ancient herpesviruses, bacteria, and old pathogens such as hepatitis virus B and C, and influenza[39]. The TRBV genes encode part of the beta chain, which, together with the alpha chain (encoded by TRAV), form the T cell receptor's antigen binding site. Notably, 8 TRBV but no TRAV genes are identified. SIRPG is a signal-regulatory protein (SIRP) member and is involved in the negative regulation of receptor tyrosine kinase-coupled signalling processes. It affects the signal regulatory protein gamma (SIRPγ) expression on T-cells and helps regulate immune responses, cell adhesion, and phagocytosis. IL20RA mediates the pro-inflammatory effects of IL-20 cytokines, helps to regulate immune responses, tissue homoeostasis, and inflammation, and is a central player in the immune system. TMPRSS11E affects epithelial barrier function, inflammation and wound healing. Conversely, only two genes out of the 16 with only an AAS signal are associated with the immune system, as the CNR2 gene product has anti-inflammatory effects, among other non-immune related functions, and PDK1 is a key regulator of immune cell development and function.

## Discussion

Local ancestry inference is fundamental to understanding the genetic history of admixed populations, and fundamentally all populations are admixed. Our study presents efficient tools for performing ancestry inference that substantially enhance computational efficiency while retaining inference accuracy. This achievement comes from the observation that in large panels, relatively few matches are required to describe local ancestry, even in the presence of sequencing error, facilitating fine-scale haplotype analyses for large-scale projects that aim to paint thousands or even millions of individuals, such as the UK Biobank and the larger biobanks of the future.

Our tools are extensions of chromosome painting to describe genome-wide ancestries, and are not specifically designed for local ancestry inference. SparsePainter extends the use of Chromosome Painting[18] for learning complex admixture histories[1] to large (10k+) reference panels[33] and it is the optimal tool for this purpose. Simulations show that it is also the best tool for fine-scale (i.e. many populations) local ancestry estimation. However, if there are few populations, FLARE or RFMix may offer edges in terms of speed, memory, or accuracy. When reference panels are themselves very admixed, MOSAIC offers a two-stage HMM that allows the reconstruction of ancestries from imperfect reference panels. Conversely, the efficiency of PBWTpaint for genome-wide ancestry estimation under reference-vs-reference painting makes it inaccurate at the level of local ancestry.

This work's broad implications extend beyond just technical improvement. The HCs computed using PBWTpaint allow robust prediction of PCs and may capture subtle genetic variations that PCs

overlook, e.g. we found improved birthplace prediction performance within the UK Biobank. Haplotype summaries have other desirable properties such as not being associated with particular genomic regions, so replacing PCs with HCs is likely to result in a similar improvement as with ancestry components (ACs)[33], which require comparison to a reference panel as SparsePainter is designed for. We therefore left a thorough examination of this task to future work and focused on the visualisation of population genetic structure using HCs at scale.

We presented a more in-depth exploration of two measures of selection at the ancestry level - LDAS which identifies ancestry segments that are too short (or too long), and AAS which identifies regions with unusual ancestry patterns. We have been careful to treat these as 'putative selection' when interpreting them because there are other reasons for these anomalies to occur. LDAS and AAS would be sensitive to SNP density, long repeats, regions with many low-quality reads, or other structural issues. AAS is particularly sensitive to the makeup of the reference panel, which must be 'less admixed' than the target individuals on average to obtain a signal. LDAS is also sensitive to recombination map details (though the recombination rate for each ethnicity is separately normalised). Although (as we have attempted) such issues are typically removed in quality control or by post-hoc considerations (low data volume regresses to the prior genome-wide ancestry), we know of no other methods that can confirm these types of selection on this timescale.

AAS has previously linked infection in admixed Scottish wildcat Felis silvestris to selectively retain an immune response developed in domestic cats Felis catus[40] over just 10 generations. Here, without looking specifically for it, we found many strong signals for core immune genes for all ethnicities using LDAS and AAS signals in the UK Biobank, which can be explained if there was a change in selection when these modern populations were formed as a mixture from older populations. Dating each would be very valuable - the admixture is only hundreds of years old for the African and European admixture seen in Afro-Caribbeans, and the last few thousand years for established populations described by 26 inter-continental populations from the 1000 Genomes Project. This historical timescale is consistent with the continued expansion of populations and their pathogens around the globe and implies a 'melting pot' of diverse diseases that evolved locally, likely related to environmental and cultural factors[41] and spread into global impact. For example, two selected immune genes (MRC1 and STAM) which have higher South Asian ancestries than expected facilitate the entry of the dengue virus, which is estimated to have evolved approximately 500-1000 years ago and first became endemic in parts of South and South-East Asia[42,43]. Today, it is widespread globally, and its range continues to expand as global warming increases the mosquito habitat that carries the dengue virus. It remains to be seen if the signal we see is this or some older virus that affects a related immune response.

It is hard to obtain ground truthing for selection statistics, and LDAS being relatively new and population-specific by design is no exception. We have attempted to rule out the most obvious confounders - beyond the usual quality control, we removed low or heterogeneous SNP density regions, which preferentially removes regions near centromeres, telomeres and indels, as well as testing for GC bias and structural variation (Methods). The strongest evidence is the clear interpretability of the signal as being immunity-associated in all 7 ethnicities. Additional evidence is needed before coming to firm conclusions, but we believe that this strongly motivates more widespread investigation of local ancestry outside of the reconstruction of individual and population histories.

Our analysis suggested that varying genetic selective patterns prevailed at different levels of control of a hierarchical complex biological system such as the immune system. Using these methods with carefully constructed reference panels targeting particular admixture times, and the analysis of specific contact events, could eventually build the pathogenic landscape around the world, and bring insights into more diseases and traits selected in our recent ancestors.

## Methods

### Ethics

This study used data from the UK Biobank (application number 80499), and from the 1000 Genomes Project, which adhered to ethical standards and guidelines established for human genome research. The UK Biobank has obtained ethical approval from the North West Multi-centre Research Ethics Committee (MREC). The 1000 Genomes Project provides publicly available genetic data that do not require additional institutional review board (IRB) approval for secondary analysis. As this study exclusively used previously approved and publicly available datasets without direct involvement of human participants, separate IRB approval was not required. The study was designed and conducted in compliance with all the relevant regulations regarding the use of human study participants and the criteria set by the Declaration of Helsinki.

### Modes of SparsePainter and PBWTpaint

As in Fig. 1, there are three modes of SparsePainter and one mode of PBWTpaint as below. The painting with a leave-one-out strategy (as required for GLOBETROTTER[1] and related methods) is classified as panel painting, which is only possible for SparsePainter.

(1) all-vs-all. Under this mode, we paint each individual against all the other individuals, i.e. only the individual itself is left out. This is for clustering, computing HCs, or similar tasks. PBWTpaint has the best performance of speed and can only operate in this mode.

(2) reference-vs-reference painting with $npop$ populations. For exchangeability between a target and the reference, one individual is left out of each other population and oneself is left out from the own population. Then we paint a reference panel against itself. This 'panel painting' makes a palette for each of the $npop$ populations as required for GLOBETROTTER[1], NNLS, and related admixture estimation methods. SparsePainter is efficient for this.

(3) target-vs-reference with $npop$ populations. We paint target individuals using a reference panel. We can optionally use leave-one-out painting (one individual is left out of each population) for admixture estimation, or without leave-one-out, we can do local ancestry inference. SparsePainter is efficient for this.

### Algorithm 'ReportQLongestMatches'

The code implementation of the PBWT structure in SparsePainter drew extensively from Sanaullah et al.[24]. We extend the 'long match query' algorithm of PBWT in Algorithm 'ReportQLongestMatches' which aims to find at least $Q$ longest matches at each position for a target sample $i$, in a two-stage process. In stage 1 we ensure a minimum number of matches, by storing only matches of length $L_{min}$ or longer containing SNPs with fewer than $Q$ matches in a set $\{s\}$. For efficiency, we search the longest matches first, by iteratively halving the match length $L_q$, beginning from $L_0$. For every SNP that still has fewer than $Q$ matches, all matches longer than $L_q$ containing the SNP are added to $\{s\}$, until all positions have at least $Q$ matches or the halved length falls below $L_{min}$.

Stage 2 reduces the number of matches. First, we calculate the genetic length for each match in $\{s\}$ and sort them in descending order of their genetic lengths. An empty set $\{e\} = \emptyset$ is then populated with only the required matches. The algorithm traverses through the sorted $\{s\}$, adding a match to $\{e\}$ if any of its positions have fewer than $Q$ matches in $\{e\}$. The final set $\{e\}$, containing elements that each specify the start position, end position, and reference sample number, represents the selected long matches to the reference haplotypes for the target sample $i$.

**Algorithm 1**. ReportQLongestMatches—find at least $Q$ Longest matches at each position for target sample $i$

 Stage 1: Ensuring a minimum number of matches;
 Run PBWT and record all matches longer than or equal to $L_0$ SNPs in set $\{s\}$.
 Let $\mathbf{r}$ be a list of SNP indices with fewer than $Q$ matches;
 Iteration $q \leftarrow 1$ and current minimum length $L_q \leftarrow L_0/2$;
 **while** $|\mathbf{r}| \neq 0 \lor L_q \geq L_{min}$ **do**
 Run PBWT and with minimum length $L_q$;
 Add matches containing SNPs in $\mathbf{r}$ with length $\in [L_q, L_{q-1})$ to set $\{s\}$;
 Update $\mathbf{r}$ with the indices of SNPs with fewer than $Q$ matches of length $L_q$ or longer;
 Half the minimum match length $L_q$ subject to constraints, i.e. $L_{q+1} \leftarrow \max(L_q/2, L_{min})$;
 $q \leftarrow q + 1$;
 **end while**
 Stage 2: Reduce the number of matches;
 ▷ *Retain only the longest, required matches.*
 Compute the genetic distance of each match in set $\{s\}$ and store in $\{g\}$
 Sort set $\{s\}$ in descending order of $\{g\}$;
 Define $\{e\}$ as an empty set to record final selected matches;
 **for** $b \leftarrow 1$ to $|s|$ **do**
 Add match $s[b]$ to set $\{e\}$ if it contains SNPs with fewer than $Q$ matches;
 If all SNPs have at least $Q$ matches **break**;
 **end for**
 Report $\{e\}$ as the selected long matches to reference haplotypes for target sample $i$.

The efficiency of this algorithm is reflected by the majority of the genome being processed in Stage 1 with few long matches, even though there are huge numbers of matches throughout the genome. Subsequently, we only need to proceed to search relatively short sections of genomes for few relatively short matches.

Note that because of the limitation of $L_{min}$, we may end up having fewer than $Q$ matches or even no matches at specific positions. The former doesn't decrease the accuracy of local ancestry inference, and we will address the latter in Methods – hidden Markov model.

## Hidden Markov model in vector form

Let $N$ be the number of haplotypes in the reference panel $K$ be the number of SNPs, and $\mu$ be the mutation probability per SNP. $\lambda$ is a recombination scaling constant, proportional to effective population size in simple demographies and called $N_e$ in Lawson et al.[18]. The reference panel $X$ is an $N$ by $K$ matrix, and a target haplotype $y$ is an $K$-vector, all taking values of either 0 or 1 corresponding to whether the reference allele is present or not. However, we can simplify this into a match matrix $M$ of dimension $N \times K$ which also takes values of either 0 or 1, with $M_{ij} = 1$ if $X_{ij} = y_j$ and 0 otherwise. We will refer to the row vectors $\mathbf{m}_j = M_j$ and use the shorthand $D(\mathbf{x}) = \text{Diag}(\mathbf{x})$ as the matrix with the vector $\mathbf{x}$ on the diagonal. We will refer to $D_N(x)$ as an $N \times N$ matrix with the scalar $x$ on the diagonal.

SparsePainter implements the Li and Stephens model[17] in the form of ChromoPainter[18] in a sparse setting. We define $\mathbf{V}$ as the emission matrix, and the column vectors are $\mathbf{v}_j = V_{\cdot j}$

$$\mathbf{V_{ij}} = \begin{cases} 1 - \mu & \text{if } \mathbf{M}_{ij} = 1 \\ \mu & \text{if } \mathbf{M}_{ij} = 0 \end{cases} \tag{1}$$

The observation matrix is an $N \times N$ matrix:

$$\mathbf{O}_j = (1 - \mu)D_N(\mathbf{m}_j) + \mu D_N(\mathbf{1}_N - \mathbf{m}_j) = D_N(\mathbf{v}_j) \tag{2}$$

The transition matrix from position $j$ to position $j + 1$ is an $N \times N$ matrix:

$$\mathbf{T}_j = \rho_j D_N(1) + \frac{1 - \rho_j}{N}\mathbf{1}_{N \times N} \tag{3}$$

where $\rho_j = \exp(-\lambda g_j)$ with $g_j$ being the genetic distance between position $j$ and position $j + 1$ in Morgans.

Let $f_0 = 1/N$ be the prior probabilities for the matches. We can write the forward probabilities for $j = 1, ..., K$ as

$$\mathbf{f}_j = \mathbf{f}_{j-1}\mathbf{T}_{j-1}\mathbf{O}_j, \tag{4}$$

where $\mathbf{f}_j$ are row vectors $(1 \times N)$. With $\mathbf{b}_K = \mathbf{1}_N$ where $\mathbf{1}_N$ is an $1 \times N$ row vector, the backward probabilities for $j = 1, ..., K - 1$ are

$$\mathbf{b}_j^T = \mathbf{T}_j\mathbf{O}_{j+1}\mathbf{b}_{j+1}^T. \tag{5}$$

However, Equation (4) and (5) can be significantly simplified due to the special form of the output and transition matrices. We can arrive at a **vector form** for which computations are $O(N)$ instead of $O(N^2)$.

To simplify notation, we write the marginal (partial) probabilities $\sum_{i=1}^N f_{ij} = \tilde{f}_j$ and $\sum_{i=1}^N b_{ij} = \tilde{b}_j$, the total number of matches $\tilde{m}_j = \sum_{i=1}^N m_{ij}$, and $\tilde{\rho}_j = \frac{1-\rho_j}{N}$. These are all scalar properties in what follows below. For the forward probabilities:

$$\begin{aligned} \mathbf{f}_j &= \mathbf{f}_{j-1}[\rho_{j-1}D_N(1) + \tilde{\rho}_{j-1}\mathbf{1}_{N \times N}][(1-\mu)D(\mathbf{m}_j) + \mu D(\mathbf{1}_N - \mathbf{m}_j)] \\ &= \mathbf{f}_{j-1}\rho_{j-1}[(1-\mu)D(\mathbf{m}_j) + \mu D(\mathbf{1}_N - \mathbf{m}_j)] + \tilde{f}_{j-1}\tilde{\rho}_{j-1}\mathbf{1}_N[(1-\mu)D(\mathbf{m}_j) + \mu D(\mathbf{1}_N - \mathbf{m}_j)] \\ &= (1-\mu)\mathbf{m}_j \circ [\rho_{j-1}\mathbf{f}_{j-1} + \tilde{f}_{j-1}\tilde{\rho}_{j-1}\mathbf{1}_N] + \mu[\rho_{j-1}\mathbf{f}_{j-1} + \tilde{f}_{j-1}\tilde{\rho}_{j-1}\mathbf{1}_N] \circ (\mathbf{1}_N - \mathbf{m}_j) \\ &= \mathbf{v}_j \circ [\rho_{j-1}\mathbf{f}_{j-1} + \tilde{f}_{j-1}\tilde{\rho}_{j-1}\mathbf{1}_N] \end{aligned} \tag{6}$$

where we use the notation $\mathbf{x} \circ \mathbf{y}$ for entry-wise vector multiplication (Hadamard product). Similarly for the backward probabilities, using the shorthand $\mathbf{c}_j = \mathbf{m}_j \circ \mathbf{b}_j$ and $\sum_{i=1}^N m_{ij}b_{ij} = \tilde{c}_j$:

$$\begin{aligned} \mathbf{b}_j^T &= [\rho_j D_N(1) + \tilde{\rho}_j\mathbf{1}_{N \times N}][(1-\mu)D_N(\mathbf{m}_{j+1}) + \mu D_N(\mathbf{1}_N - \mathbf{m}_{j+1})]\mathbf{b}_{j+1}^T \\ &= \rho_j[(1-\mu)D_N(\mathbf{m}_{j+1}) + \mu D_N(\mathbf{1}_N - \mathbf{m}_{j+1})]\mathbf{b}_{j+1}^T \\ &\quad + \tilde{\rho}_j\mathbf{1}_{N \times N}[(1-\mu)(\mathbf{m}_{j+1} \circ \mathbf{b}_{j+1})^T + \mu \mathbf{b}_{j+1}^T \circ (\mathbf{1}_N - \mathbf{m}_{j+1})^T] \\ &= \rho_j(1-\mu)\mathbf{c}_{j+1}^T + \rho_j\mu(\mathbf{b}_{j+1}^T - \mathbf{c}_{j+1}^T) + \tilde{\rho}_j(1-\mu)\tilde{c}_{j+1}\mathbf{1}_N^T + \tilde{\rho}_j\mu(\tilde{b}_{j+1} - \tilde{c}_{j+1})\mathbf{1}_N^T \\ &= \rho_j(\mathbf{c}_{j+1}^T - 2\mu\mathbf{c}_{j+1}^T + \mu\mathbf{b}_{j+1}^T) + \tilde{\rho}_j(\tilde{c}_{j+1} - 2\mu\tilde{c}_{j+1} + \mu\tilde{b}_{j+1})\mathbf{1}_N^T \\ &= \rho_j\mathbf{d}_{j+1}^T + \tilde{\rho}_j\tilde{d}_{j+1}\mathbf{1}_N^T \end{aligned} \tag{7}$$

where $\mathbf{d}_j = \mathbf{v}_j \circ \mathbf{b}_j$ and such that $\sum_{i=1}^N v_{ij}b_{ij} = \tilde{d}_j$. Finally, the posterior probabilities are written in the following form:

$$P(\mathbf{m}_j|\mathbf{O}) \propto \mathbf{f}_j \circ \mathbf{b}_j^T. \tag{8}$$

If we assume the mutation rate $\mu \to 0$, the forward and backward probabilities (Equation (6) and (7)) simplify to

$$\mathbf{f}_j = \mathbf{m}_j \circ \left[\rho_{j-1}\mathbf{f}_{j-1} + \tilde{f}_{j-1}\tilde{\rho}_{j-1}\mathbf{1}_N\right] \tag{9}$$

and

$$\mathbf{b}_j^T = \rho_j\boldsymbol{\delta}_{j+1}^T + \tilde{\rho}_j\tilde{\delta}_{j+1}\mathbf{1}_N^T \tag{10}$$

respectively, where $\boldsymbol{\delta}_j = \mathbf{m}_j \circ \mathbf{b}_j$ and $\tilde{\delta}_j = \sum_{i=1}^N m_{ij}b_{ij}$. In this case, only the forward probabilities $\mathbf{f}_j$ for the matched samples at position $j$ are non-zero and need to be calculated. For backward probabilities, we compute different $\mathbf{b}_j^T$ for matched samples at position $j + 1$, with unmatched samples sharing the same default value $\tilde{\rho}_j\tilde{\delta}_{j+1}$ in the $j$th hash vector. Finally, when computing the posterior probabilities $P(\mathbf{m}_j|\mathbf{O})$ (Eq. (8)), only samples with matches in SNP $j$ or $j + 1$ require computation, whereas the others are exactly 0.

Note that this assigns non-zero probability to single mutation breaks in haplotypes, provided a match is found both to the left and the right. In conclusion, the Hash-Map-based forward and backward algorithm reduces computational cost from $O(N)$ (e.g., ChromoPainter[18]) to approximately $O(Q)$.

There are instances when few positions have no matches spanning at least $L_{min}$ SNPs, and are therefore interpreted as no matches, which disrupts the forward and backward algorithm because a 0-vector of $\mathbf{f}_j$ causes all $\mathbf{f}_t$ to become 0-vectors for any $t > j$. To address this issue, for each position without matches longer than $L_{min}$ SNPs, we find the closest SNP (in genetic distance) that has matches. We then impute the matches from this closest SNP to the position without matches.

The recombination scaling constant $\lambda$ is usually estimated by the Expectation-Maximisation (E-M) algorithm (Supplementary Methods). However, the Viterbi algorithm, a dynamic programming technique to identify the most probable sequence of hidden states in a hidden Markov model, can be advantageously employed to improve the efficiency of estimating $\lambda$, compared with the E-M algorithm. In this context, let $N_{seg}$ represent the minimum number of contiguous segments from different reference samples required to construct the target haplotype, and therefore $N_{break} = N_{seg} - 1$ is essentially the number of distinct recombination events that have been inferred. Then $\lambda$ is estimated as

$$\lambda^* = \frac{N_{break}}{\sum_{j=1}^{K} g_j}. \tag{11}$$

### The normalised versions of the forward and backward equations

It is helpful to work in the normalised versions of the forward and backward equations $\check{\mathbf{f}}_j = \mathbf{f}_j/\check{f}_j$ and $\check{\mathbf{b}}_j = \mathbf{b}_j/\check{b}_j$. We define $F_j$ and $B_j$ as the normalising constant at state $j$.

$$\frac{\mathbf{f}_j}{\check{f}_{j-1}} = \mathbf{m}_j \circ \left[(1-\mu)\left(\rho_{j-1}\check{\mathbf{f}}_{j-1} + \tilde{\rho}_{j-1}\mathbf{1}_N\right) - \mu\left(\rho_{j-1}\check{\mathbf{f}}_{j-1} + \tilde{\rho}_{j-1}\mathbf{1}_N\right)\right] + \mu\left[\rho_{j-1}\check{\mathbf{f}}_{j-1} + \tilde{\rho}_{j-1}\mathbf{1}_N\right] \tag{12}$$

Setting $\mu \to 0$, $\mathbf{v}_j$ shrinks to $\mathbf{m}_j$:

$$\begin{aligned} \mathbf{f}_j &= \mathbf{m}_j \circ \left[\rho_{j-1}\mathbf{f}_{j-1} + \check{f}_{j-1}\tilde{\rho}_{j-1}\mathbf{1}_N\right] \\ \check{\mathbf{f}}_j &= \frac{\check{f}_{j-1}}{\check{f}_j}\frac{\mathbf{f}_j}{\check{f}_{j-1}} = \frac{\check{f}_{j-1}}{\check{f}_j}\mathbf{m}_j \circ \left[\rho_{j-1}\check{\mathbf{f}}_{j-1} + \tilde{\rho}_{j-1}\mathbf{1}_N\right] \\ &= \frac{1}{F_j}\mathbf{m}_j \circ \left[\rho_{j-1}\check{\mathbf{f}}_{j-1} + \tilde{\rho}_{j-1}\mathbf{1}_N\right] \end{aligned} \tag{13}$$

which has the following consequences:

 (a) Let $s_j$ be the set of matches at SNP $j$: $i \in s_j \Leftrightarrow m_{ij} = 1$.
 (b) $\check{f}_{ij}^* = \rho_{j-1}\check{f}_{i(j-1)} + \tilde{\rho}_{j-1}$ if $i \in s_j$ and is zero otherwise.
 (c) $F_j = \sum_{i \in s_j} \check{f}_{ij}^*$ and $\check{f}_{ij} = \check{f}_{ij}^*/F_j$.
 (d) for a sparse algorithm, we only need to track matches and the relative sums of their probabilities.

For the backward algorithm with $\mu \to 0$, $\mathbf{d}_j$ shrinks to $\mathbf{c}_j$:

$$\begin{aligned} \mathbf{b}_j^T &= \rho_j \mathbf{c}_{j+1}^T + \tilde{\rho}_j \check{c}_{j+1}\mathbf{1}_N^T \\ \check{\mathbf{b}}_j^T &= \frac{\check{c}_{j+1}}{\check{b}_j}\left[\rho_j \check{\mathbf{c}}_{j+1}^T + \tilde{\rho}_j \mathbf{1}_N^T\right] \end{aligned} \tag{14}$$

which has the following consequences:

 (a) $\check{b}_{ij}^* = \rho_j \check{b}_{i(j+1)} + \tilde{\rho}_j \check{c}_{j+1}$ if $i \in s_{j+1}$ and $\check{b}_{ij}^* = \tilde{\rho}_j \check{c}_{j+1}$ otherwise, where $\check{c}_{j+1} = \sum_{i \in s_{j+1}} \check{b}_{i(j+1)}$.
 (b) $B_j = \sum_{i \in s_{j+1}} \check{b}_{ij}^* + (N - n_{j+1})\tilde{\rho}_j \check{c}_{j+1}$ and $\check{b}_{ij} = \check{b}_{ij}^*/B_j$.
 (c) Again this can be computed without explicit reference to non-matches and we need to sum over only matches.

### Estimation of the expected length of copied chunks

Let $\hat{l}_i$ denote the posterior expected length (in Morgans) of the total genome for which the sample haplotype copies from the $i$th reference haplotype.

$$\begin{aligned} \hat{l}_i &= \frac{1}{2\Pr(D)}\sum_{j=1}^{K-1} g_j \left[f_{ij}b_{ij} + f_{i(j+1)}b_{i(j+1)}\right] \\ &= \frac{1}{2\prod_{k=1}^{K}F_k}\sum_{j=1}^{K-1} g_j \left[\check{f}_{ij}\check{b}_{ij}\left(\prod_{k=1}^{j}F_k\right)\left(\prod_{k=j}^{K}B_k\right) \right. \\ &\quad \left. + \check{f}_{i(j+1)}\check{b}_{i(j+1)}\left(\prod_{k=1}^{j+1}F_k\right)\left(\prod_{k=j+1}^{K}B_k\right)\right] \\ &= \frac{1}{2}\sum_{j=1}^{K-1} g_j \left[w_j^l \check{f}_{ij}\check{b}_{ij} + w_j^r \check{f}_{i(j+1)}\check{b}_{i(j+1)}\right] \end{aligned} \tag{15}$$

where

$$w_j^l = \exp\left(\log\left(\prod_{k=1}^{j}F_k\right) + \log\left(\prod_{k=j}^{K}B_k\right) - \log\left(\prod_{k=1}^{K}F_k\right)\right)$$

and

$$w_j^r = \exp\left(\log\left(\prod_{k=1}^{j+1}F_k\right) + \log\left(\prod_{k=j+1}^{K}B_k\right) - \log\left(\prod_{k=1}^{K}F_k\right)\right).$$

### Estimation of the expected number of chunks copied

Let $\hat{c}_i$ denote the posterior expected number of chunks copied from the $i$th reference haplotype.

$$\begin{aligned} \hat{c}_i &= \frac{1}{\Pr(D)}f_{i1}b_{i1} + \frac{1}{\Pr(D)}\sum_{j=1}^{K-1}\left[f_{i(j+1)}b_{i(j+1)} - f_{ij}b_{i(j+1)}V_{i(j+1)}\rho_j\right] \\ &= \frac{1}{\prod_{k=1}^{K}F_k}\check{f}_{i1}\check{b}_{i1}F_1 B_1 \\ &\quad + \frac{1}{\prod_{k=1}^{K}F_k}\sum_{j=1}^{K-1}\left[\check{f}_{i(j+1)}\check{b}_{i(j+1)}\left(\prod_{k=1}^{j+1}F_k\right)\left(\prod_{k=j+1}^{K}B_k\right) \right. \\ &\quad \left. - \check{f}_{ij}\check{b}_{i(j+1)}\left(\prod_{j=1}^{j}F_j\right)\left(\prod_{k=j+1}^{K}B_k\right)V_{i(j+1)}\rho_j\right] \\ &= \frac{1}{\prod_{k=1}^{K}F_k}\check{f}_{i1}\check{b}_{i1}F_1 B_1 + \sum_{j=1}^{K-1}\left[a_j^l\check{f}_{i(j+1)}\check{b}_{i(j+1)} - a_j^r\check{f}_{ij}\check{b}_{i(j+1)}V_{i(j+1)}\rho_j\right] \end{aligned} \tag{16}$$

where

$$a_j^l = \exp\left(\log\left(\prod_{k=1}^{j+1}F_k\right) + \log\left(\prod_{k=j+1}^{K}B_k\right) - \log\left(\prod_{k=1}^{K}F_k\right)\right)$$

and

$$a_j^r = \exp\left(\log\left(\prod_{k=1}^{j}F_k\right) + \log\left(\prod_{k=j+1}^{K}B_k\right) - \log\left(\prod_{k=1}^{K}F_k\right)\right).$$

### Non-negative least squares (NNLS) for admixture estimation

Admixture estimation can be performed on both the reference individuals and the target individuals via NNLS, which requires the expected total genome shared between each reference ancestry, and

each reference (or target) individual with each reference ancestry. The former is derived by painting the reference samples against themselves with one sample left out of each other population (i.e. reference-vs-reference painting). We then average the expected length of copied chunks for each reference individual within each reference ancestry to provide a reference palette. When investigating admixture estimation for target individuals, we also require painting each target sample (i.e. target-vs-reference painting) against a reference panel, with one sample left out from every reference ancestry. Reference (target) samples are then described as a mixture of the reference ancestries using NNLS, calculated by the R package 'nnls'[1]. In detail, we fit the NNLS model by minimising $\|Ax - b\|_2$ with the constraints $x \geq 0$, where $A$ is the reference palette and $b$ is the expected length of copied chunks for each reference (target) sample, and finally obtain the estimates $x$.

## Simulation details for comparison between SparsePainter, PBWTpaint, ChromoPainter, FLARE, RFMix and MOSAIC

We simulated different simple models (Simulation 2a-c) for target-vs-reference painting, and a hierarchical model (Simulation 1) for reference-vs-reference painting. Each simulation is repeated 10 times, and the average statistics, i.e. compute time, memory usage and accuracy, are reported.

The simple simulation model for target-vs-reference painting (Simulation 2a-c) begins with an ancestral population of 50000 individuals that evolved for 2500 generations prior to diverging into $n_{pop}$ populations with different sizes. Here we specify the population sizes for Simulation 2b-c, including $n_{pop}$ = 2, 3, 5:

(1) 5000 and 20000 for 2-way admixture model;
(2) 5000, 15000 and 25000 for 3-way admixture model;
(3) 5000, 10000, 7000, 14000 and 9000 for 5-way admixture model.

Following an evolutionary period of another 500 generations, these $n_{pop}$ populations admixed into 1000 modern individuals with different proportions. Again, we specify the admixture proportions for Simulation 2b-c:

(1) 50% and 50% for 2-way admixture model;
(2) 20%, 50% and 30% for 3-way admixture model;
(3) 20%, 10%, 10%, 40% and 20% for 5-way admixture model.

The admixed individuals had a growth rate of 5% per generation, and they were sampled 13 generations after admixture.

For Simulation 1, we constructed a hierarchical model that mirrors the evolutionary trajectory of real-world populations, which is used for the comparison of reference-vs-reference panel painting. We simulated a 5-population and a 10-population hierarchical model. Here we illustrate the 5-population ($P_i(i = 1, 2, ..., 5)$) model in detail. After an ancestral population with 10000 individuals evolved for 2700 generations, it split into $P_1$ and $P_4$ with 7000 and 3000 individuals. After generation 2890, $P_2$ emerged from migrations originating from $P_1$ with a population size of 3000. Moving forward to the 2940th generation, 1000 people from $P_2$ migrated to a new population $P_3$. A final migration occurred at the 2950th generation when 2000 individuals from $P_4$ settled to create $P_5$. All the populations had a growth rate of 5% from the 2970th to the 3000th generation. At the 3000th generation, we sampled an equivalent number of individuals (20, 40, 60, 100, 180, 300 and 500) from each population $P_i(i = 1, 2, ..., 5)$. A similar model was constructed for simulating 10 hierarchical populations.

Because all methods considered are linear in genome length, all simulations (Simulation 1 and Simulation 2a-c) use 20 megabases (Mb) of genome, characterised by a mutation rate of $1.44 \times 10^{-8}$ per base pair per generation, a recombination rate of $1 \times 10^{-8}$ Morgans per base pair per generation. Following Browning et al.[20], we included gene conversion at twice the recombination rate with an average tract length of 300 base pairs, and genotype error with a proportion of 0.02%. We

retained 20k SNPs with Minor Allele Frequency (MAF) ≥1% shared between the reference and target datasets.

The true local ancestry is defined as 5 generations before admixture, which is derived from the recombination events recorded in the tree sequences (in SLiM) during the 500 generations before admixture. Some regions (around 10%–20%) in target haplotypes were inherited from the ancestral population and haven't experienced any recombination events during the 500 generations. As in Browning et al.[20], to compare the local ancestry estimates we excluded the SNPs within those regions, but we emphasise that the 'true' local ancestry of these regions can only be defined in terms of a mixture of the descendent populations. Genome-wide ancestry estimates are obtained by summing the probabilities as in ChromoPainter[18].

For all the simulations, we also retained 20,000 common SNPs with MAF of at least 1% from the reference and target datasets presented in the Variant Call Format (VCF). In detail, all simulations generated more than 20,000 SNPs after MAF filtering, and we sampled 20,000 random SNPs from them for analysis. Subsequently, we merged the reference and target datasets and phased the merged dataset with Beagle 5.4[44] before splitting it into the reference and target datasets. FLARE requires input data in VCF format, while ChromoPainter requires phase format, and SparsePainter and PBWTpaint enable both input formats (we used the phase format). Phase format can be efficiently converted from VCF with PBWT.

For SparsePainter, unless otherwise stated, we ensured no more than 10 longest matches (longer than 20 SNPs) at each locus are retained. All simulations are performed on an MSI laptop with an Intel Core i7-10750H processor running at 2.60GHz on 10 CPU cores in parallel.

We explored a number of different parameters for Simulation 2a-c.

(1) Simulation 2a: we simulated 2-, 5-, 10-, 20-, 40-, 60-, 80- and 100-way admixture ($npop$ = 2, 5, 10, 20, 40, 50, 80 and 100) models to compare the speed and memory of painting 50 admixed individuals between software, with varying numbers of total reference sizes (2000, 4000 and 8000) with random numbers of (at least 10) individuals per reference ancestry.

(2) Simulation 2b: we simulated 2-, 3- and 5-way admixture ($n_{pop}$ = 2, 3 and 5) models to compare the local ancestry inference accuracy of 50 admixed individuals between software, with varying numbers of reference sizes for each reference ancestry (100, 200, 500, 1000 and 2000).

(3) Simulation 2c: we drew from reference pools of 1000, 2000, or 4000 individuals for each of the $n_{pop}$ = 5 reference ancestries. We then evaluated SparsePainter's efficiency in painting 1000 admixed individuals under varying levels of sparsity, i.e. only the longest 5, 10, 20, 40 and 80 matches which are longer than 20 SNPs are retained at each SNP. This was manipulated via the 'nmatch' parameter in SparsePainter.

## Methods to evaluate the accuracy of local ancestry and NNLS estimates

We used two different methods to assess the accuracy of local ancestry estimates. The first method is the squared Pearson's correlation coefficient (denoted as $r^2$). At each SNP, we calculated the estimated dosage of each individual by averaging the posterior probabilities of both haplotypes for each reference ancestry, and the true dosage is the average true local ancestry which takes values of 0, 0.5, or 1. We computed the $r^2$ between the estimated and actual dosages for each reference ancestry across all individuals and positions, and the unweighted mean $r^2$ of these values is reported to measure the overall accuracy. The second method evaluates the proportion of accurate local ancestry predictions across all haplotypes and positions. For each haplotype at a specific position, a

correct local ancestry inference is determined when the true local ancestry corresponds to the highest estimated posterior probability, i.e. the best-guess strategy.

To evaluate the accuracy of admixture estimation, we calculated the squared correlation between the NNLS-estimated coefficient (see above) and the true proportion for all the individuals, and reported the unweighted mean $r^2$ of NNLS from different populations.

## The accuracy of PBWTpaint and using set-maximal matches for local ancestry estimation

Unlike SparsePainter, PBWTpaint does not provide a calibrated estimate of local ancestry. To assess this, we compare local ancestry estimates under 5-way reference-vs-reference panel painting with 500 reference individuals per population. On the simple simulation model (Simulation 2a-2c) in which the populations are distinct, the $r^2$ between PBWTpaint and SparsePainter is 0.79. However, for complex cases in which there is uncertainty, or the true ancestry is an ancestor of extant populations (Simulation 1), the set maximal matches used by PBWTpaint lead to over-confident or inaccurate local ancestry assignment ($r^2 = 0.3$) even though these mistakes are self-averaging for the estimation of genome-wide ancestry. This illustrates that PBWTpaint is not an appropriate method for performing local ancestry estimates.

We also compared the accuracy of using set-maximal matches with the Li and Stephens HMM for local ancestry inference, under 5-way reference-vs-reference panel painting with 500 samples per population. In detail, we obtained the set-maximal matches via the 'maxWithin' command of PBWT, and then input these matches with '-matchfile' in SparsePainter to calculate the local ancestry probabilities for each reference individual. We used exactly the same model as above, and found that $r^2$ between SparsePainter LAI with set-maximal matches and $Q$ longest matches (the default SparsePainter algorithm) is 0.89 for simple simulation model with distinct populations (Simulation 2a-2c) but only 0.72 under complex cases (Simulation 1). Therefore, using set-maximal matches is not suitable for accurate LAI.

## Methods to evaluate the accuracy of SparsePainter under different genotyping errors, small reference sizes, different phasing errors, high recombination rates, and different admixture times

Under the 3-way admixture model of Simulation 2b with 500 reference individuals in each population, we evaluated the accuracy of SparsePainter compared with ChromoPainter under different genotyping errors, small reference sizes, different phasing errors, and high recombination rates. We also compared the accuracy of SparsePainter, ChromoPainter and RFMix for target individuals sampled under different generations since admixture. We used the same parameters as Simulation 2b except for the parameter that is under investigation. The accuracy in terms of the proportion of correct LAI and Pearson's $r^2$ are reported in Supplementary Tables 2–6.

## Paint all UK Biobank individuals against themselves and calculate haplotype principal components

To infer the haplotype principal components, we painted UK Biobank individuals against themselves, i.e. all-vs-all painting. We first excluded withdrawn individuals, and excluded related individuals, i.e. we ensured no relatives within 3 generations were included in the analysis as quantified by the pairwise genetic relatedness score using the '-genome' command in plink[45]. We then performed PBWTpaint (with command pbwt '-paintSparse') on each chromosome of UK Biobank phased genotype data, which in total has 406,733 individuals with 569,200 SNPs. The total chunk length of PBWTpaint for each individual on chromosome $i$ is $2K_i$, where $K_i$ is the number of SNPs. Assume $g_i$ is the total genetic distance for chromosome $i$, we weighted the chunk length for chromosome $i$ with weight $g_i/K_i$. Then we summed up the sparse chunk length matrix for all the chromosomes as matrix $A$, such that for each individual (i.e. each row of $A$), the expected lengths of copied chunks from all other individuals reached the sum of the total genetic distance $G = \sum_{i=1}^{22} g_i$.

We performed singular value decomposition (SVD) on the log-transformed sparse chunk length matrix $log10(A + 1)$ with R package 'sparsesvd': $log10(A + 1) = UDV^T$, where $D$ is a diagonal matrix of the singular values. Then we extracted the the first 150 columns of $U\sqrt{D}$ as the top 150 haplotype principal components.

## Prediction of birth locations with HCs and PCs

We conducted an analysis to evaluate the predictive accuracy of HCs and PCs on the birth locations, i.e. the east and north coordinates, within the UK. We selected a cohort of 347,532 individuals who were born in the UK or Ireland and identified as white, British, or Irish ethnicity, including both males and females. This cohort was divided into two groups: a training set comprising 80% of the individuals, and a test set consisting of the remaining 20%. Subsequently, with either the top 150 HCs or PCs as explanatory variables and either the east or north coordinate as the response variable, we used a 5-fold CV to determine the optimal number of boosting iterations before fitting the regression model on the training set with eXtreme Gradient Boosting (XGBoost[46]), and then we predicted the birth coordinates of individuals in the test set. After repeating the process for 100 times, we computed the direct distance between the predicted coordinates and the actual coordinates of each individual on the test set, and reported the median which reveals that using HCs as predictors (median error=39.7km) reduced 49% error compared with using PCs as predictors (median error=77.4km). This indicates a notably higher predictive accuracy of birthplaces when using HCs.

We also benchmarked the accuracy of HCs and PCs in predicting the population labels through the hierarchical simulation model with 500 individuals in each of the $npop = 10$ populations (Simulation 1). After computing the top 150 HCs and PCs, we sampled 80% individuals from each population as the training set, and used the remaining individuals as the test set. Similarly, we used a 5-fold CV to determine the optimal number of boosting iterations before fitting the classification model on the training set with XGBoost, and then predicted the population labels of the individuals in the test set. We repeated the simulation 100 times using different numbers of HCs and PCs (5, 10, 20, 50, 100 or 150), and finally reported the average and standard deviation of the proportion of accurate prediction of population labels in Supplementary Table 7. The results clearly demonstrate that HCs reach almost 100% accuracy with only 5 HCs used, and consistently achieve much stronger association with genome-wide ancestry compared to PCs.

## Paint UK Biobank with 1000 Genomes Project

We inferred the local ancestry of UK Biobank individuals using the 1000 Genomes Project as the reference data, which includes 2504 individuals (both males and females) from 26 populations. We retained the common bi-allelic SNPs with MAF ≥ 5% before merging these two datasets. Then we used Beagle 5.4[44] to phase the merged dataset, after which it was split into the reference and target datasets. For a comparative analysis of the genetic painting and population structure within the UK Biobank, we randomly selected 10,000 individuals (both males and females) with self-reported British backgrounds, and incorporated all individuals from specific self-reported ethnic backgrounds: Irish (12713), Indian (5660), Caribbean (4297), African (3203), Pakistani (1747), and Chinese (1503).

We estimated the average recombination scaling constant $\lambda = 164.2$ of all these individuals on chromosome 19. This fixed parameter was subsequently used for painting across chromosomes 1-22. We configured the parameters of SparsePainter to aim for finding the 50 longest matches (longer than 20 SNPs) at each position.

**Quality control for shared and ethnicity-specific LDAS and AAS**

Here we explain the method for finding shared and ethnicity-specific LDAS and AAS, and the additional Quality Controls (QC) applied. As introduced by Barrie et al.[12], we compute the LDAS of SNP $j$ in principle as the integral of the LDA between every other position genome with $g_j$, over the recombination map with length $L_j$ consisting of the chromosome holding the $j$-th SNP:

$$LDAS(j) = \int_0^{L_j} LDA(g, g_j)\, dg. \tag{17}$$

In practice, the pairwise LDA shrinks to almost 0 when the closest SNPs are more than 3 centiMorgan (cM) away, so the integral is approximated over a $X = 4$cM window as LDAS$(j; X)$ by:

$$LDAS(j; X) = \begin{cases} \int_{g_j-X}^{g_j+X} LDA(g, g_j)\, dg & \text{if } X \le g_j \le L_j - X, \\ \int_0^{g_j+X} LDA(g, g_j)\, dg + \int_{2g_j}^{g_j+X} LDA(g, g_j)\, dg & \text{if } g_j < X, \\ \int_{g_j-X}^{L_j} LDA(g, g_j)\, dg + \int_{g_j-X}^{2g_j-L_j} LDA(g, g_j)\, dg & \text{if } g_j > L_j - X. \end{cases} \tag{18}$$

where $g_j$ is the genetic position in centiMorgan for the $j$th SNP, and LDA$(g, g_j)$ is the LDA between position $g$ and the target SNP at position $g_j$.

Because LDA can only be computed at discrete SNPs, in practice these integrals are approximated, which leads to an error that must be controlled. If the SNPs present are random with respect to the true recombination locations, then the lowest mean-square-error estimate of LDAS in Equation (18) integral treats LDA as a piecewise linear function:

$$LDA(g, g_j) = (1 - \alpha)LDA(g_i, g_j) + \alpha LDA(g_{i+1}, g_j),$$

where $\alpha = (g - g_i)/(g_{i+1} - g_i) \in [0, 1)$ for $g \in [g_i, g_{i+1}]$. Further, an upper bound and lower bound can be obtained by replacing the piecewise linear function with a step function. In detail, we take the larger and smaller LDA values of two neighbouring SNPs, respectively, as the fixed LDA in the genetic distance between two SNPs $i$ and $j$ in the integral:

$$LDA_{upper}(g, g_j) = \max\left\{ LDA(g_i, g_j), LDA(g_{i+1}, g_j) \right\}$$

and

$$LDA_{lower}(g, g_j) = \min\left\{ LDA(g_i, g_j), LDA(g_{i+1}, g_j) \right\}.$$

These estimates are substituted into Equation (18) to obtain an upper and lower bound respectively of the LDAS of SNP $j$. When computing LDAS$_{lower}(j; X)$, we assume the chromosome ends have zero LDA with the target SNP, i.e. LDA$(0, g_j) = $ LDA$(L_j, g_j) = 0$ for conservative estimation.

The maximum possible error in the LDAS estimate at SNP $j$ is

$$LDAS_{error}(j; X) = LDAS_{upper}(j; X) - LDAS_{lower}(j; X). \tag{19}$$

It is necessary to account for different scales of LDAS across different ethnic backgrounds, because of different admixture times with respect to the populations in the panel. Therefore, for each ethnic background, we standardise the LDAS$_{error}$ with the average LDAS across the genome, i.e. LDAS$^*_{error} = $ LDAS$_{error}/E(LDAS)$. Finally, in the QC we remove SNPs with large relative error, i.e. LDAS$^*_{error} \ge \delta$ where $\delta$ is a specified threshold (we used $\delta = 0.3$). This provides an implicit condition of high SNP density with respect to the recombination map.

A final challenge is that no LDA can be detected if SNPs are very sparse, so that LDAS$_{upper}$ is estimated near zero and the error is undefined. We therefore remove SNPs if any nearby 0.5cM region

within 3cM has too few SNPs: SNP $j$ is removed if at least one of $n_m(j) < \theta$ for $m = 0.5, 1, 1.5, 2, 2.5, 3$, where $n_m(j)$ is the number of SNPs that is $(m - 0.5, m]$cM away from SNP $j$ and $\theta$ is a specified threshold (we used $\theta = 10$).

In conclusion, we use two additional filters; firstly that LDAS$_{error} < \delta$ and $n_m \ge \theta$ ($m = 0, 0.5, 1.0, 1.5, 2, 2.5$) as the quality control of SNPs, which alleviates the biased estimates due to sparsity of the painting data and therefore avoids unreliable LDA scores. In practice, this removes 3.5% of the genome (20,075 out of 569,200 SNPs) in 62 contiguous segments (see Supplementary Data 1–7 for details). Because of the SNP selection process inherent in the UK Biobank genotyping chip, these are predominantly centromeres, telomeres, and regions that already have SNPs removed due to standard QC procedures, including where there are missing data due to e.g. indels, alignment issues, etc.

The computation of AAS is not affected by the discrepancy of recombination events across chromosomes and ethnicities, and we implemented the procedures as described in Barrie et al.[12] with SparsePainter.

As validated through simulation, we assume the normality of LDAS for all ethnicities across the genome. We converted the LDAS into p-values through the one-sided normality test which aims to detect low LDAS, and we only focused on SNPs with LDAS from at least one ethnic background that is significant at $p = 10^{-6}$. Those SNPs are classified as shared or ethnicity-specific low LDAS if LDAS from all the other ethnic backgrounds are significant at $p = 0.05$, or insignificant at $p = 0.1$, respectively.

As AAS approximately follows a Gamma distribution and produces more significant p-values (through the one-sided Gamma test), we employed a stricter significance level, $p = 10^{-50}$, for filtering SNPs with significant AAS. Similarly, those SNPs are categorised as having shared or ethnicity-specific significant AAS if AAS from all the other ethnic backgrounds are significant at $p = 10^{-10}$, or insignificant at $p = 10^{-5}$, respectively.

Furthermore, to ensure robust results, we repainted UKB using 5 continental populations as delineated by the 1000GP continents (Europe, Africa, America, South Asia and East Asia) to obtain an alternate set of LDAS and AAS results. We then mapped each SNP with low LDAS and AAS signals to its gene (if the SNP overlaps with a gene) via R package 'gprofiler2', and visualised the results in Figs. 5, 6.

To ensure the validity of LDAS and AAS signals, we evaluated their association with GC bias. Using the GC frequency reported for East Asia, Europe and Africa[47], we found that all the regions with shared LDAS or AAS signals had random frequencies of G+C (Supplementary Fig. 10), which showed no evidence of association with GC bias. We also checked the association of LDAS and AAS signals with structural variation (SV). We downloaded the regions with SV in 1000GP[48], and found this covers 3.91% of the whole genome. For SNPs with LDAS or AAS signals, we classified them into various small regions which are no longer than 10kb, and computed the proportion of these regions that have SV. We detected 7.45%, 7.04%, 7.48% and 8.89% of the regions with 26-pop LDAS, 5-continent LDAS, 26-pop AAS, and 5-continent AAS signals, respectively. Whilst it is plausible that selection acts on SV and LDAS jointly, we cannot rule out reverse causation of SV causing an LDAS or AAS signal without selection. Therefore those regions with SV were excluded from further analysis, though this choice does not materially affect the conclusions (see Supplementary Data 8 for a list of SNPs affected).

**Simulation for LDAS under genetic drift**

We assessed the robustness of the LDAS and its sensitivity to demographic changes by examining it under genetic drift across exponentially expanding population sizes over time. We simulated a genome of a 500Mb region as follows: initially, an ancient population evolves for 1000 generations, subsequently diverging into five

distinct subpopulations. Each of these subpopulations, growing at a rate of 2% per generation, evolves independently for 100 generations. This period of divergence is followed by a phase of admixture, forming a modern, unified population, which then undergoes evolution for an additional 30 generations at an increased growth rate of 5% per generation.

We computed the LDAS of 500 simulated modern individuals with 2000 simulated reference individuals from each of the 5 subpopulations. After standardisation, the z-scores of the LDAS (Supplementary Fig. 9a) predominantly exhibit under-dispersion, despite some noticeable deviation on both tails. This pattern suggests that the normal distribution is a reasonable approximation for the LDAS distribution. Subsequently, we calculated the p-values for low LDAS through a one-sided test for normality, as depicted in Supplementary Fig. 9b. Notably, no low LDAS signals are detected under the genetic drift model (excluding selection effects), as evidenced by the most significant SNP with $p < 10^{-3}$ through the one-sided normality test. This outcome solidifies our conclusion that low LDAS signals are not present under this model.

### Simulation for comparing LDAS with statistics for positive selection

Here we simulated the similar two-loci and one-locus model as used in Barrie et al.[12].

For the two-loci selection model (Supplementary Fig. 6), we simulated a genome of 150Mb. Initially, an ancient population evolved for 2200 generations before splitting into two sub-populations $P1$ and $P2$. After evolving 400 generations, we added mutation $m1$ for $P1$ and $m2$ for $P2$ at locus 20Mb and 23Mb, respectively. These added mutations were then positively selected in the following 300 generations before admixing to $P3$ at generation 2900. $m1$ and $m2$ then experienced strong positive selection for another 50 generations, after which we sampled 500 individuals from $P3$ as target individuals. 500 individuals are sampled for $P1$ and $P2$ at generation 2899 as the reference panel.

For the one-locus selection model (Supplementary Fig. 7), we simulated a genome of 50Mb. The remaining difference from the above mode is that only one locus $m0$ at 20Mb was added at generation 2601 for both $P1$ and $P2$, and it was positively selected until generation 2900. In the admixture population $P3$, this SNP underwent negative selection until generation 2950 when the target individuals were sampled.

Both simulations had a mutation rate of $1.44 \times 10^{-8}$ per base pair per generation, and a recombination rate of $1 \times 10^{-8}$ Morgans per base pair per generation.

### Comparison of LDAS and AAS signals with natural selection in Bronze Age Britain and archaic adaptive introgression in 1000GP populations

Our LDAS and AAS analyses from painting 7 UK Biobank ethnic backgrounds with 1000GP populations have detected various signals of selection (Fig. 6 and Supplementary Table 1), and we investigated the overlaps with the other selection signals. By comparison with the genome-wide significant ($P < 10^{-7}$) selection signals in the ancient British data[49], we found the only overlap genes are HLA-DRB6 and HLA-DRB1 on chromosome 6. We compared loci that have been identified as exhibiting adaptive introgression from Neanderthal or Denisovan ancestries in the 1000GP populations[50]. Although none of them overlaps genes with LDAS signals, we discovered that the ADARB2 gene, located on chromosome 10 overlaps with AAS signals. This gene experiences introgression from Denisovan ancestry within the 1000GP PEL population, and coincides with the AAS signals in British, Irish, Indian, Caribbean and Chinese ethnicities. Notably, the utilisation of different reference panels can probably lead to the identification of distinct genes exhibiting selection signals of LDAS and AAS.

### Reporting summary

Further information on research design is available in the Nature Portfolio Reporting Summary linked to this article.

## Data availability

The phased 1000 Genomes Project data build GRCh37/hg19 are available at https://bochet.gcc.biostat.washington.edu/beagle/1000_Genomes_phase3_v5a/b37.vcf/. The genetic map data build GRCh37/hg19 are available at https://bochet.gcc.biostat.washington.edu/beagle/genetic_maps/. The UK Biobank data can be accessed by approved researchers through https://www.ukbiobank.ac.uk. We used the UK Biobank data under project 81499. The UK map data are available at https://gadm.org and through R package 'rworldmap' at https://cran.r-project.org/web/packages/rworldmap/index.html. Source data are provided with this paper.

## Code availability

The C++ code for SparsePainter is available on GitHub at https://github.com/YaolingYang/SparsePainter[26] (https://doi.org/10.5281/zenodo.14640171), and the website for SparsePainter is at https://sparsepainter.github.io/. PBWTpaint is available on GitHub at https://github.com/richarddurbin/pbwt[25] (https://doi.org/10.5281/zenodo.14753968). The UK Biobank painting pipeline and methods to compute haplotype components (HCs) are available on GitHub at https://github.com/YaolingYang/SparsePainter/tree/main/painting-pipeline.

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

## Acknowledgements

The authors thank the participants in the UK Biobank and 1000 Genomes Project. Y.Y. was supported by China Scholarship Council [grant number 202108060092]. A.K.N.I. was supported by the OAK foundation (OFIL-20-095). This work was carried out using the computational facilities of the Advanced Computing Research Centre, University of Bristol - http://www.bris.ac.uk/acrc.

## Author contributions

Y.Y., R.D. and D.J.L. conceived and designed the project and methodology. D.J.L. supervised the project. Y.Y., R.D. and D.J.L. developed the methodology. R.D. and D.J.L. programmed the codes for PBWTpaint. Y.Y. programmed the codes for SparsePainter. Y.Y. did simulations and UKB data analysis under the supervision of D.J.L. A.K.N.I. analysed and interpreted genes with LDAS and AAS signals. Y.Y. wrote the initial manuscript draft. All authors wrote, reviewed, discussed and revised the subsequent versions of the manuscript (led by Y.Y. and D.J.L.). Y.Y. and A.K.N.I. wrote the Supplementary Information. All authors agreed with the submitted manuscript.

## Competing interests

The authors declare no competing interests.
