## [Transparent Peer Review file · Nature Communications]

Sparse haplotype-based fine-scale local ancestry inference at scale reveals recent selection on immune responses

Corresponding Author: Mr Yaoling Yang

Version 0:

Reviewer comments:

Reviewer #1

(Remarks to the Author)

This paper introduced a local ancestry inference method, SparsePainter. Additionally, it introduced a global ancestry estimation method, PBWtpaint. Both methods used the Positional Burrows-Wheeler Transform (PBWT) algorithm to extract matches as intermediate information, then used the Li and Stephens (LS) model or match overlapping estimation for ancestry inference. The authors argued that the proposed methods were faster than existing ones and had comparable accuracy. Overall, this is a well-written paper with good presentation and readability. However, the novelty of the methods should be better claimed, and the significance of the contribution should be clearly stated.

The authors claimed the proposed methods were novel because they harnessed the strength of PBWT which made them run fast. However, some existing ancestry methods already adopt PBWT such as FLARE (the authors cited in this paper) which uses PBWT to find matches from the reference panel and build composite haplotypes from them as the intermediate data. A detailed explanation is needed about how PBWT was used differently than the way that FLARE did and why it was innovative.

The LAI results of the proposed methods were comparable to existing LAI methods, but no significant accuracy improvement. If the goal of this study was to show the proposed methods helped improve the compute time drastically while maintaining high accuracy, how to keep the balance between compute time and accuracy should be discussed. For example, an experiment can be conducted to show when SparsePainter reaches at what sparsity level its ancestry inference accuracy would start decreasing, and how the selection of the Q value would affect the result.

More explanation is needed about why a simple model using set-maximal matches only is not suitable for estimating local ancestries (line 635). It is not mentioned if the poor performance was caused by the match overlapping estimation that PBWtpaint used or the set-maximal matches that it used. It is possible to have a target-vs-reference enabled version of PBWtpaint so it can be compared to SparsePainter directly in a local ancestry inference experiment. In the PBWT paper (<https://doi.org/10.1093/bioinformatics/btu014>), Durbin described an algorithm (section 2.4 Finding all set-maximal matches from a new sequence z to X) that finds set-maximal matches between a target and a panel. It would be interesting to see the comparison between them. Also, another possible test is to have a version of PBWtpaint using set-maximal matches with the LS model like SparsePainter does. This should tell if using the same LS model, which type of matches is better for LAI, set-maximal matches, or long matches.

The authors included PBWtpaint (this study), SparsePainter (this study), and ChromoPainter (Lawson et al., 2012) for the performance comparison on genome-wide ancestry inference. The result showed that PBWtpaint is the fastest among the methods. Some recently published methods should also be included for the comparison. For example, Neural ADMIXTURE (Dominguez Mantes et al., 2023) (<https://doi.org/10.1038/s43588-023-00482-7>), a neural network-based ADMIXTURE version was proposed recently, which was claimed that “the compute time was reduced by orders of magnitude surpassing even the fastest alternatives”. Or at least some justifications are needed to explain why including ChromoPainter which was published 12 years ago only is sufficient for this experiment.

It was not clear if SparsePainter was designed for local ancestry inference on target individuals from the most recent admixture events. The experiments seem to only cover the case of recent admixture events. Based on the simulation setup, the admixed individuals were sampled 13 generations after admixture (line 564). Using the Q longest matches may be

enough for the case. Those long matches may be generated by recent recombination events. However, the length of the matches found may be much shorter when admixture events occur in a distant time (e.g., 100 generations ago). Additional experiments may be conducted to show how SparsePainter's performance is, or at least a discussion about the relationship between match length and the number of generations after admixture events is needed.

Since both set-maximal matches and long matches from PBWT are exact matches, the impact of errors in the dataset should be investigated. Having large genotyping errors in the dataset may break the long matches into a few short matches, which may impact the performance of the proposed methods. It seems the performance of the proposed methods was good if there was a 0.02% genotyping error (line 581). It is good for the potential users to know approximately how many genotyping errors in the dataset that the proposed methods can tolerate.

(Remarks on code availability)

Reviewer #2

(Remarks to the Author)

(Remarks on code availability)

Reviewer #3

(Remarks to the Author)

This work presents two near-linear time algorithms to infer ancestry in large genetic data using the Positional Burrows-Wheeler Transform (PBWT). One of the algorithms, SparsePainter, uses a set of haplotypes and HMM to infer local ancestries. The other algorithm, PBWtpaint, uses only the longest set-maximal which has performance in terms of run time and memory. The authors compared their algorithms against ChromoPainter and FLARE. The results show that the methods may perform better in some cases but they are necessarily producing a lot more accurate results. Also, methods such as FLARE are scalable which reduces the significance of the presented algorithm.

- The presented algorithms are well-described. Besides the fact that separating the two algorithms causes some confusion at the beginning for the reader. However, my main criticism is the significance of the work. While the presented algorithms may be faster (or less resource-intensive), other available methods such as FARE can be used in large cohorts. The following points may help improve the manuscript and showcase the significance of the work.

- For the admix population, MOSAIC may perform better than other tools but its performance has not been shown here. Also, RFMIX has been omitted which can perform better in some cases. Based on Figure 3, it is hard to make a conclusion about which tool performs better. While no tools may have superior performance in all cases, more details are missing on where and why SparsePainter has better accuracy in some simulations.

- Haplotype Principal Components analysis has been performed on UKBB and compared against PCA. However, it lacks proper benchmarking via simulation to show the advantages of using HC vs. PCA (such as better association with birthplace).

(Remarks on code availability)

Reviewer #4

(Remarks to the Author)

The authors present two novel methods for chromosome painting and local ancestry inference, both leveraging the PBWT data structure to efficiently identify large matches between haplotypes. Their application to the UK Biobank is commendable and demonstrates the utility of SparsePainter.

While the methods show notable improvements over existing approaches, particularly in terms of scalability and applicability to large datasets, the paper would benefit from clearer presentation. Additionally, the benchmarking lacks sufficient depth and clarity to fully support the claims made.

Comments:

A1. SparsePainter introduces approximations to the Li and Stephens model, including 1) exclusion of reference haplotypes with short matches, 2) omission of shorter matches within considered haplotypes, and 3) a zero mutation rate, which limits the set of haplotypes updated during forward-backward calculations. Although the authors show with simulations that these approximations do not significantly affect accuracy, it remains unclear if these assumptions hold in more challenging scenarios. The benchmark would benefit from further evaluation in conditions such as small reference panels, phasing errors, complex admixture, or high recombination regions. This would strengthen claims of "negligible accuracy drop."

A2. The approximation of the Li and Stephens model is innovative, but it resembles FLARE's approach, which involves haplotype selection followed by composite reference panel creation. The authors should address the computational distinctions between SparsePainter and FLARE. Additionally, FLARE's quadratic complexity with increasing number of populations is interesting, and a discussion of why SparsePainter maintains constant complexity in this context would be valuable.

A3. The benchmarking and simulation sections are somewhat difficult to follow. The reference panel sizes in each experiment are unclear, as experiments are continuously mixed though the benchmark (e.g., Figures 2 and 3).

A4. Given SparsePainter's scalability focus, I expected benchmarks using much larger reference panels, potentially in the range of hundreds of thousands or millions of haplotypes. While SparsePainter's performance is impressive, it remains uncertain whether FLARE might be more efficient for very large reference panels with a moderate number of populations (e.g., ≤ 30).

A5. A benchmark with real datasets (e.g., 1KGP, HDGP) would provide additional insights and help illustrate the practical differences between the methods.

A6. The manuscript's first part could benefit from a more clear rewriting, and some figures do not effectively highlight the key points. For example, Figure 1 might be more effectively presented as a concise table. Similarly, Figure 6 could also be condensed in a table. Also, due to the complexity of presenting two similar methods, a simplified visual representation of the methods—especially SparsePainter—could clarify their differences for readers less interested in the technical details. The method section, however, is well-written and easier to follow.

Minor comments:

B1. Figures 2 and 3 are difficult to interpret and would benefit of full re-plotting. The dataset used should be more clearly indicated within the figures, and its naming could be more meaningful.

B2. The rationale behind selecting specific simulations should be clearer in the main text, with a better description of what each simulation aims to achieve.

B3. ChromoPainter has quadratic complexity with the size of the reference. However, Figures 2c and 2e show a slightly more than linear relationship in time and memory usage, but not quadratic. So a phrase such as "fig. 2a-b illustrates that both in theory and practice, ChromoPainter has a quadratic cost as a function of panel size, so scales poorly to larger reference sizes" might need rephrasing.

B4. Some wording issues appear in the manuscript:

Page 4, L116: "maga-scale" should likely be "large-scale."

Page 4, L123: Two outputs are mentioned, but only one is explained.

Page 5, L155: This introduction also applies to PBWTpaint and might be better placed earlier.

Page 6, L175: "Stephen's" should be corrected.

B5. The GitHub page for the software is well-organized, and the authors have made the tool accessible and user-friendly.

(Remarks on code availability)

Version 1:

Reviewer comments:

Reviewer #1

(Remarks to the Author)

The authors have addressed most of our concerns in this revision. However, there are still two comments remaining.

In response to the question “how PBWT was used differently than the way that FLARE did and why it was innovative”, the authors stated that “SparsePainter also uses PBWT technically different from FLARE: SparsePainter uses ‘ReportQLongestMatches’ to ensure the sparsity of matches, which is an improvement to the original PBWT algorithm ‘ReportLongMatches’.” However, based on the description of the “ReportQLongestMatches” algorithm, it seems “ReportQLongestMatches” algorithm is just a few post-processing steps on the matches resulting from the original PBWT algorithm “ReportLongMatches” with different length threshold parameters. Our understanding is that both SparsePainter and FLARE use the same PBWT “ReportLongMatches” algorithm to get matches. Depending on the different length thresholds used, SparsePainter and FLARE get different sets of matches. The authors should talk about why a particular set of matches was needed for SparsePainter and how it would help to improve the local ancestry inference accuracy.

In response to the question of how SparsePainter’s performance is for local ancestry inference on target individuals from the distant admixture events, the authors stated that “We have explored and found that all the LAI methods (including SparsePainter, FLARE, RFMix and ChromoPainter) have significantly decreasing accuracy with more distant admixture time”. However, this statement or any benchmarking results supporting this statement were not found in the manuscript or supplementary material.

(Remarks on code availability)

Reviewer #2

(Remarks to the Author)

(Remarks on code availability)

Reviewer #3

(Remarks to the Author)

The authors have responded to all the issues I raised in the initial submission.

(Remarks on code availability)

PBWTpaint GitHub repo does not have proper README or instructions on how to use it.

Reviewer #4

(Remarks to the Author)

The paper presents a method whose most noteworthy results are its scalability and ability to handle a large number of reference populations. These strengths are now validated through a wider range of simulations conducted under more challenging, non-ideal conditions.

While restructuring the paper could enhance readability and impact, I leave this decision to the authors and editor.

I have no further comments.

(Remarks on code availability)

Version 2:

Reviewer comments:

Reviewer #1

(Remarks to the Author)

The authors have addressed our concerns in this revision.

(Remarks on code availability)

Reviewer #2

(Remarks to the Author)

I co-reviewed this manuscript with one of the reviewers who provided the listed reports. This is part of the Nature

Communications initiative to facilitate training in peer review and to provide appropriate recognition for Early Career Researchers who co-review manuscripts.

(Remarks on code availability)

Reviewer #1 (Remarks to the Author):

This paper introduced a local ancestry inference method, SparsePainter. Additionally, it introduced a global ancestry estimation method, PBWtpaint. Both methods used the Positional Burrows-Wheeler Transform (PBWT) algorithm to extract matches as intermediate information, then used the Li and Stephens (LS) model or match overlapping estimation for ancestry inference. The authors argued that the proposed methods were faster than existing ones and had comparable accuracy. Overall, this is a well-written paper with good presentation and readability. However, the novelty of the methods should be better claimed, and the significance of the contribution should be clearly stated.

The authors claimed the proposed methods were novel because they harnessed the strength of PBWT which made them run fast. However, some existing ancestry methods already adopt PBWT such as FLARE (the authors cited in this paper) which uses PBWT to find matches from the reference panel and build composite haplotypes from them as the intermediate data. A detailed explanation is needed about how PBWT was used differently than the way that FLARE did and why it was innovative.

Response: Thank you for pointing out this. The way we and FLARE use PBWT (and the Li & Stephens model) is quite different, as seen in e.g. the scaling properties of the two algorithms. FLARE uses PBWT to find long matches between reference and target samples for IBD estimation. As such FLARE doesn't assign ancestry to every part of the genome (as we describe below). It solves conceptually different problems from our algorithms, which use PBWT for chromosome painting, i.e. assigning each genome segment ancestry (as introduced by Lawson et al., 2012) and which can be used in complex admixture history modelling (Hellenthal et al., 2014). While FLARE is useful in relatedness patterns instead of ancestry assignment, PBWtpaint is faster than FLARE because it requires minimal post-processing of the pbwt output, simply computing a weighting based on the length distribution at a site to replicate ChromoPainter.

SparsePainter also uses PBWT technically different from FLARE: SparsePainter uses "ReportQLongestMatches" to ensure the sparsity of matches, which is an improvement to the original PBWT algorithm "ReportLongMatches".

Changes: We have updated Fig. 1 caption to better capture this, and updated our explanatory paragraph (Line 130): "Chromosome Painting is conceptually different to identifying haplotypes identical-by-descent (IBD); it assigns every position of the genome to the most-recent common ancestor in the reference..."

The LAI results of the proposed methods were comparable to existing LAI methods, but no significant accuracy improvement. If the goal of this study was to show the proposed methods helped improve the compute time drastically while maintaining high accuracy, how to keep the balance between compute time and accuracy should be discussed. For example, an experiment can be conducted to show when SparsePainter reaches at what

sparsity level its ancestry inference accuracy would start decreasing, and how the selection of the Q value would affect the result.

Response: Thank you for suggesting additional discussion here. We have investigated the trade-off between accuracy and compute time (and also memory usage) by using different sparsity level (controlled by Q ranging from 5 to 80 under specific total reference sizes), as shown in Fig. 3c-d. From the plot we find that using Q=20 would make the accuracy close to the “optimal” while staying quite efficient in speed and memory, and larger reference samples dilute the accuracy's sensitivity to sparsity. The exact value of Q would also be affected by other factors such as the density of SNPs. We have provided additional discussion on this.

Changes: We have added the following paragraph in the “Results” section (Line 266-269): “Using Q=20 matches saturates accuracy whilst minimising speed and memory cost, and larger reference samples dilute the accuracy's sensitivity to sparsity. However, SNP density may influence the exact Q value needed for optimal performance.”

More explanation is needed about why a simple model using set-maximal matches only is not suitable for estimating local ancestries (line 635). It is not mentioned if the poor performance was caused by the match overlapping estimation that PBWTpaint used or the set-maximal matches that it used. It is possible to have a target-vs-reference enabled version of PBWTpaint so it can be compared to SparsePainter directly in a local ancestry inference experiment. In the PBWT paper (<https://doi.org/10.1093/bioinformatics/btu014>), Durbin described an algorithm (section 2.4 Finding all set-maximal matches from a new sequence z to X) that finds set-maximal matches between a target and a panel. It would be interesting to see the comparison between them. Also, another possible test is to have a version of PBWTpaint using set-maximal matches with the LS model like SparsePainter does. This should tell if using the same LS model, which type of matches is better for LAI, set-maximal matches, or long matches.

Response: Thank you for this valuable comment. We used simulations to show that both the PBWTpaint algorithm and using set-maximal matches cause the poor LAI estimation of PBWTpaint.

We agree that this is possible in principle. Unfortunately, in the “Methods” section “The accuracy of PBWTpaint and using set-maximal matches for local ancestry estimation” we found that the local ancestry inference of PBWTpaint is extremely low under complex admixture scenarios. Given this inaccuracy, we feel that the effort to develop target-vs-reference in PBWTpaint for local ancestry inference is of lower priority (in other words, the additional modeling done by the Li & Stephens model is adding value).

Use set-maximal matches from PBWTpaint for LAI using the LS model is possible through ‘-matchfile’ option in SparsePainter. We have used the same simulation to

compare this and presented the results in the same method section “The accuracy of PBWTpaint and using set-maximal matches for local ancestry estimation”, which indicates the inaccuracy of LAI using set-maximal matches.

Changes: We have added the following contents in the “Methods” section (Line 671-679): “We also compared the accuracy of using set-maximal matches with the Li and Stephens HMM for local ancestry inference, under 5-way reference-vs-reference panel painting with 500 samples per population. In detail, we obtained the set-maximal matches via the ``maxWithin`` command of PBWT, and then input these matches with ``-matchfile`` in SparsePainter to calculate the local ancestry probabilities for each reference individual. We used exactly the same model as above, and found that r^2 between SparsePainter LAI with set-maximal matches and Q longest matches (the default SparsePainter algorithm) is 0.89 for simple simulation model (Simulation 2a-2c) but only 0.72 under complex cases (Simulation 1). Therefore, using set-maximal matches is not suitable for accurate LAI.”

The authors included PBWTpaint (this study), SparsePainter (this study), and ChromoPainter (Lawson et al., 2012) for the performance comparison on genome-wide ancestry inference. The result showed that PBWTpaint is the fastest among the methods. Some recently published methods should also be included for the comparison. For example, Neural ADMIXTURE (Dominguez Mantes et al., 2023) (<https://doi.org/10.1038/s43588-023-00482-7>), a neural network-based ADMIXTURE version was proposed recently, which was claimed that “the compute time was reduced by orders of magnitude surpassing even the fastest alternatives”. Or at least some justifications are needed to explain why including ChromoPainter which was published 12 years ago only is sufficient for this experiment.

Response: Thank you for suggesting the comparison with Neural ADMIXTURE, which due to its near linear-time compute is an interesting competitor. Initially we had included ChromoPainter in our benchmarking analysis because SparsePainter updates the ChromoPainter framework and a direct comparison is interesting. As suggested, we included Neural ADMIXTURE in our benchmarking study, and found that PBWTpaint is still much faster and requires much less memory than Neural ADMIXTURE.

The comparison is not direct because Neural ADMIXTURE is designed to estimate admixture proportions directly. It does not estimate expected proportion of genome most recently inherited from a population, or local ancestry, so we cannot appropriately compare the accuracy. However, Neural ADMIXTURE uses SNP data, and not haplotypes or IBD, and we therefore can rely on a vast body of literature showing that these genomic features are much more informative especially over recent timescales (Lawson et al., 2012, Hellenthal et al., 2014, Pagani et al., 2016, Browning et al., 2023, Speidel et al., 2018, Hu et al., 2023, amongst many more). Tellingly, Neural ADMIXTURE was not compared to these techniques, presumably because of this mismatch of purpose. Since the genome-wide accuracy of PBWTpaint is comparable to ChromoPainter under complex simulation models, it is fair to conclude that PBWTpaint is

more efficient at genome-wide ancestry inference. (The new Supplementary Table 6, generated in response to Reviewer #3 below, provides additional evidence that SNPs simply lack information that is present in haplotypes for this task, as does Fig. 4. Many papers, e.g. McVean 2009, describe the close mathematical relationship between PCA and Admixture; Lawson & Falush 2012 discusses this.)

Changes: We have updated the benchmarking with Neural ADMIXTURE in Fig. 2 and the simulation subsection in the “Results” section (Line 207-221).

It was not clear if SparsePainter was designed for local ancestry inference on target individuals from the most recent admixture events. The experiments seem to only cover the case of recent admixture events. Based on the simulation setup, the admixed individuals were sampled 13 generations after admixture (line 564). Using the Q longest matches may be enough for the case. Those long matches may be generated by recent recombination events. However, the length of the matches found may be much shorter when admixture events occur in a distant time (e.g., 100 generations ago). Additional experiments may be conducted to show how SparsePainter’s performance is, or at least a discussion about the relationship between match length and the number of generations after admixture events is needed.

Response: Thank you for suggesting the investigation of SparsePainter’s performance with more distant admixture events, and for identifying that we didn’t fully explain the use cases for SparsePainter. We have explored and found that all the LAI methods (including SparsePainter, FLARE, RFMix and ChromoPainter) have significantly decreasing accuracy with more distant admixture time.

In all cases, the match length reduces with more distant admixture time. The most natural use case for SparsePainter is to investigate LAI on target individuals from recent admixture events. Also, for more distant admixture events, SparsePainter is still useful to provide estimates for segments of DNA that were inherited from old events by treating the match length distribution as a mixture, as done using GLOBETROTTER (Hellenthal et al., 2014) using ChromoPainter as input. SparsePainter is a direct, but more efficient, functional replacement for this pipeline, allowing much larger sample sizes and therefore increased accuracy for a given compute.

Additionally, with more distant admixture time, although SparsePainter could become inaccurate for LAI, it is accurate for global ancestry estimation (as indicated by Fig. 3a, i.e. high accuracy under complex admixture models), which is one of the motivating use cases for chromosome painting, and why IBD approaches (i.e. FLARE) do not report results for all of the genome.

Changes: When introducing the purpose of SparsePainter, we have added (Line 126-129) “The second is the expected fraction of the total genome shared most recently between a target and each reference ancestral individual or population, as used in complex admixture history modelling, e.g. the GLOBETROTTER tool.”

Since both set-maximal matches and long matches from PBWT are exact matches, the impact of errors in the dataset should be investigated. Having large genotyping errors in the dataset may break the long matches into a few short matches, which may impact the performance of the proposed methods. It seems the performance of the proposed methods was good if there was a 0.02% genotyping error (line 581). It is good for the potential users to know approximately how many genotyping errors in the dataset that the proposed methods can tolerate.

Response: Thank you for pointing out our inadequate discussion and investigation on the degree of genotyping error that SparsePainter can tolerate. We performed simulation to simulate different genotyping errors in both reference and target samples (0.02%, 0.05%, 0.1%, 0.2%, and 0.5%) under 3-way admixture model between SparsePainter and ChromoPainter. We initially also compared with FLARE, but we found that FLARE can only compute IBD for very few SNPs under high genotyping error (around 80% SNPs when genotyping error=0.02%, but only <1% SNPs when genotyping error=0.5%) because of its algorithm, so it was removed from comparison.

We showed the result in Supplementary Table 2 which indicates that SparsePainter can tolerate the genotyping error around 0.1%-0.2%. SNP array data has around 0.1% genotyping error (Saunders et al., 2007, modern platforms may be slightly better), so therefore we highlight that SparsePainter is suitable for either SNP array data or carefully curated medium-high sequence depth whole genome sequencing data, i.e. strict quality control should be needed in order to use SparsePainter.

We also showed through simulation that the accuracy of SparsePainter does not have significant difference from that of ChromoPainter under different phasing error, recombination rate, and small reference sizes, following the suggestion of Reviewer #4. These results are reported in Supplementary Tables 3-5.

Changes: We have reported the results in the “Results” section (Line 258-261) “We show using simulation (Methods) that SparsePainter can tolerate genotyping error (Supplementary Table 2) up to around 0.1%-0.2%. This is superior to FLARE, and SparsePainter only has negligible accuracy loss under small reference sizes (Supplementary Table 3), different phasing errors (Supplementary Table 4), and high recombination rates (Supplementary Table 5).”

We have reported the accuracy comparison of genotyping errors in Supplementary Table 2, and written an additional subsection (Line 681-687) “Methods to evaluate the accuracy of SparsePainter under different genotyping errors, small reference sizes, different phasing errors and high recombination rates” with the following paragraph in the “Methods” section: “Under the 3-way admixture model of Simulation 2b with 500 reference individuals in each population, we evaluated the accuracy of SparsePainter under different genotyping errors, small reference sizes, different phasing errors and high recombination rates. We used the same parameters as Simulation 2b except for the parameter that is under investigation. The accuracy in terms of the proportion of correct LAI and Pearson's r^2 are reported in Supplementary Tables 2-5.”

Reviewer #2 (Remarks to the Author):

Response: Thank you for reviewing our manuscript and providing valuable suggestions.

Reviewer #3 (Remarks to the Author):

This work presents two near-linear time algorithms to infer ancestry in large genetic data using the Positional Burrows-Wheeler Transform (PBWT). One of the algorithms, SparsePainter, uses a set of haplotypes and HMM to infer local ancestries. The other algorithm, PBWtpaint, uses only the longest set-maximal which has performance in terms of run time and memory. The authors compared their algorithms against ChromoPainter and FLARE. The results show that the methods may perform better in some cases but they are necessarily producing a lot more accurate results. Also, methods such as FLARE are scalable which reduces the significance of the presented algorithm.

- The presented algorithms are well-described. Besides the fact that separating the two algorithms causes some confusion at the beginning for the reader. However, my main criticism is the significance of the work. While the presented algorithms may be faster (or less resource-intensive), other available methods such as FARE can be used in large cohorts. The following points may help improve the manuscript and showcase the significance of the work.

- For the admix population, MOSAIC may perform better than other tools but its performance has not been shown here. Also, RFMIX has been omitted which can perform better in some cases.

Response: Thank you for suggesting benchmarking against MOSAIC and RFMix. We have included the comparison of MOSAIC and RFMix in speed, memory and accuracy, and updated Fig. 2c-d and Fig. 3b, which illustrated that MOSAIC is inaccurate (because it uses an unsupervised algorithm in which reference populations are not assumed to be perfect) while RFMix is slightly more accurate than SparsePainter and FLARE under 2- and 3-way admixture but less accurate under 5-way admixture. Also, both MOSAIC and RFMix are overall more than 100 times slower than SparsePainter.

Changes: We have updated Fig. 2c-d and Fig. 3b. We have also updated the simulation section in the “Results” section (Line 223-261) with this additional benchmarking.

Based on Figure 3, it is hard to make a conclusion about which tool performs better. While no tools may have superior performance in all cases, more details are missing on where and why SparsePainter has better accuracy in some simulations.

Response: Thank you for this question. The purpose of designing SparsePainter is that we’d like to update the ChromoPainter’s framework, making SparsePainter a sparse implementation of ChromoPainter which is hundreds times faster but with comparable accuracy. We have pointed out this in Line 254 of main text that “SparsePainter is essentially a sparse implementation of ChromoPainter”, and have explained its use cases more completely when we introduce it (Line 120-129).

For comparison with FLARE, we have also shown that SparsePainter has comparable accuracy, but is much more efficient for fine-scale ancestry inference, e.g. when having more than 20 populations. Also, we notice that FLARE restricts reporting of LAI to SNPs that it can find IBD, and in practice when having even small genotyping errors (e.g. 0.05%), a very limited number of SNPs are reported. This means the reported accuracy is the upper bound for FLARE, and because of this, SparsePainter is more useful than FLARE in real situation – for example, when analysing SNP array data which has around 0.1% genotyping error (Saunders et al., 2007, modern platforms may be slightly better).

Changes: We added “Whilst the accuracy of SparsePainter and FLARE is comparable, we note that FLARE restricts reporting of LAI to SNPs that it can find IBD. The excluded SNPs have less certain LAI due to short matches, which can be caused by genotyping error or ancient admixture, so its reported accuracy is an upper bound.” in the “Results” section (Line 250-253).

- Haplotype Principal Components analysis has been performed on UKBB and compared against PCA. However, it lacks proper benchmarking via simulation to show the advantages of using HC vs. PCA (such as better association with birthplace).

Response: Thank you for your suggestion on additional simulation to show the advantages of using HCs vs PCs. Please find below the simulation with complex hierarchical model we have performed, which clearly showed that HCs have stronger association with the overall ancestry than PCs, quantified by the proportion of accurate prediction of population labels.

Changes: We have added below contents in the “Methods” section (Line 720-730) explaining the simulation, and Supplementary Table 6 which shows the simulation results: “We also benchmarked the accuracy of HCs and PCs in predicting the population labels through the hierarchical simulation model with 500 individuals in each of the $n_{pop}=10$ populations (Simulation 1). After computing the top 150 HCs and PCs,

we sampled 80% of individuals from each population as the training set, and used the remaining individuals as the test set. Similarly, we used a 5-fold CV to determine the optimal number of boosting iterations before fitting the classification model on the training set with XGBoost, and then predicted the population labels of the individuals in the test set. We repeated the simulation 100 times using different numbers of HCs and PCs (5,10,20, 50, 100 or 150), and finally reported the average and standard deviation of the proportion of accurate prediction of population labels in Supplementary Fig. 6. The results clearly demonstrate that HCs reach almost 100% accuracy with only 5 HCs used, and consistently achieve much stronger association with genome-wide ancestry compared to PCs.”

We have also added the sentence in the “Results” section (Line 299-301) “We demonstrate through simulations that only 5 HCs can almost perfectly predict genome-wide ancestries, outperforming the predictive power of even 150 PCs (see Methods and Supplementary Table 6).”

Reviewer #4 (Remarks to the Author):

The authors present two novel methods for chromosome painting and local ancestry inference, both leveraging the PBWT data structure to efficiently identify large matches between haplotypes. Their application to the UK Biobank is commendable and demonstrates the utility of SparsePainter.

While the methods show notable improvements over existing approaches, particularly in terms of scalability and applicability to large datasets, the paper would benefit from clearer presentation. Additionally, the benchmarking lacks sufficient depth and clarity to fully support the claims made.

Comments:

A1. SparsePainter introduces approximations to the Li and Stephens model, including 1) exclusion of reference haplotypes with short matches, 2) omission of shorter matches within considered haplotypes, and 3) a zero mutation rate, which limits the set of haplotypes updated during forward-backward calculations. Although the authors show with simulations that these approximations do not significantly affect accuracy, it remains unclear if these assumptions hold in more challenging scenarios. The benchmark would benefit from further evaluation in conditions such as small reference panels, phasing errors, complex admixture, or high recombination regions. This would strengthen claims of "negligible accuracy drop."

Response: Thank you for suggesting additional simulations to evaluate the accuracy profile of SparsePainter. Because SparsePainter is the sparse implementation of ChromoPainter, here we compare the accuracy between them to investigate the “accuracy drop” under a 3-way admixture simulation with 20000 SNPs and 500 reference individuals per population (except when investigating small reference panels),

and report the results in Supplementary Tables 3-5. Below is the summary of the simulations:

- (a) Small reference panels: 20,40,60,80,100 reference samples per ancestry. Both methods have lower accuracy with smaller reference panels, but the accuracy difference between SparsePainter and ChromoPainter remains similar.
- (b) Phasing errors: 0%, 0.2%, 0.5%, 1%, 1.5%, 2%. The accuracy of both methods is high, and the difference between SparsePainter and ChromoPainter remains similar.
- (c) Recombination rates: 1cM/Mb, 1.5cM/Mb, 2cM/Mb, 2.5cM/Mb 3cM/Mb. The accuracy of both methods is high, and the difference between SparsePainter and ChromoPainter remains similar.
- (d) Complex admixture: SparsePainter is not designed for LAI of complex admixture. To model complex admixture, other tools such as MOSAIC (Salter-Townshend & Myers 2019), or post-processing such as with GLOBETROTTER (Hellenthal et al., 2014) could be more appropriate.

Changes: We have reported the results in the “Results” section (Line 258-261): “We show using simulation (Methods) that SparsePainter can tolerate genotyping error (Supplementary Table 2) up to around 0.1%-0.2%. This is superior to FLARE, and SparsePainter only has negligible accuracy loss under small reference sizes (Supplementary Table 3), different phasing errors (Supplementary Table 4), and high recombination rates (Supplementary Table 5).”

We have reported the accuracy comparison of small reference panels, phasing errors, and high recombination rates in Supplementary Table 3-5, and written an additional subsection “Methods to evaluate the accuracy of SparsePainter under different genotyping errors, small reference sizes, different phasing errors and high recombination rates” with the following paragraph in the “Methods” section (Line 681-687): “Under the 3-way admixture model of Simulation 2b with 500 reference individuals in each population, we evaluated the accuracy of SparsePainter under different genotyping errors, small reference sizes, different phasing errors and high recombination rates. We used the same parameters as Simulation 2b except for the parameter that is under investigation. The accuracy in terms of the proportion of correct LAI and Pearson's r^2 are reported in Supplementary Tables 2-5.”

A2. The approximation of the Li and Stephens model is innovative, but it resembles FLARE's approach, which involves haplotype selection followed by composite reference panel creation. The authors should address the computational distinctions between SparsePainter and FLARE. Additionally, FLARE's quadratic complexity with increasing number of populations is interesting, and a discussion of why SparsePainter maintains constant complexity in this context would be valuable.

Response: Thank you for raising this point which is not inadequately discussed in the manuscript. SparsePainter (and ChromoPainter) performs ancestry-unaware painting, i.e. computes the marginal probabilities of each reference haplotype without access to labels. It then combines them according to the ancestry label, so the number of

ancestries does not increase the compute time, while the only negligible increased compute time (as shown in Fig. 2c) owing to writing results to disk. In comparison, FLARE does ancestry-aware painting which uses ancestry-specific transition rates in its hidden Markov model computation, and that results in FLARE's quadratic complexity with increasing number of populations. We have further expanded on the conceptual differences in the purpose (and hence output) of FLARE and SparsePainter (and ChromoPainter) in the response to Reviewer #1.

Changes: We have written in the "Result" section (Line 229-230): "The speed and memory of SparsePainter and ChromoPainter remain largely unaffected by the number of true populations, because both of them perform chromosome painting that is ancestry-unaware."

We have also updated our explanatory paragraph (Line 130): "Chromosome Painting is conceptually different to identifying haplotypes identical-by-descent (IBD); it assigns every position of the genome to the most-recent common ancestor in the reference..."

A3. The benchmarking and simulation sections are somewhat difficult to follow. The reference panel sizes in each experiment are unclear, as experiments are continuously mixed though the benchmark (e.g., Figures 2 and 3).

Response and Changes: To make clearer plots and write-up of the simulations, we have improved the design of Fig. 2 and Fig. 3 to make it easier to follow, i.e. the new plots clearly show the reference panel sizes, by changing to line plots it is easier to distinguish different methods. We have stated clearly each plot corresponds to which simulation in the captions of Fig. 2-3.

A4. Given SparsePainter's scalability focus, I expected benchmarks using much larger reference panels, potentially in the range of hundreds of thousands or millions of haplotypes. While SparsePainter's performance is impressive, it remains uncertain whether FLARE might be more efficient for very large reference panels with a moderate number of populations (e.g., ≤ 30).

Response: Thank you for raising the question. The purpose of SparsePainter is to step up ChromoPainter to current and future scale datasets. Whilst huge datasets are available and need to be analysed within-sample (which PBWTPaint can do), this is typically the first step in identifying the underlying populations of relatively unadmixed individuals for supervised LAI estimation.

Conversely, huge (say 1M+), accessible, reference panels of carefully labeled samples are essentially unavailable in practice - these rely on people whose ancestry via all 4 grandparents is from a specific population, who are generally rare. Further, large simulated datasets that lack realistic multi-scale complexity (hierarchical and spatial structure plus historically varying rates of admixture) are not very helpful for indicating

accuracy in reality (when reference sizes are at best moderate in scale and complex), so we decided to save the CO2 from analysing much larger simulated reference panels, as the scaling is already clear.

Notably, with larger reference panels, the only increased computational time is because of building PBWT structure for the reference panel. Through simulation 2c (Fig. 3c-d, and section “Sparsity in SparsePainter” in the main text) we investigated painting with larger reference panels (up to 20000 in total) and more individuals (1000), which shows that the computational time almost remains unchanged with increasing reference sizes.

From simulations (and Fig. 2c-d) we can conclude that FLARE is slightly more efficient when $n_{pop} \leq 5$, similar as SparsePainter when $n_{pop} = 10$, and SparsePainter is much more efficient when $n_{pop} \geq 20$. We have appropriately indicated this threshold in the “Results” section (Line 231-233): “Conversely, whilst FLARE demonstrates impressive speed and efficient memory usage with few populations ($n_{pop} \leq 5$), its efficiency dramatically diminishes compared to SparsePainter when handling 20 or more populations (Fig. 2c-d).”

A5. A benchmark with real datasets (e.g., 1KGP, HDGP) would provide additional insights and help illustrate the practical differences between the methods.

Response: This is an important point. SparsePainter is in principle a sparse implementation of ChromoPainter, which has been used in many real-data scenarios (Lawson et al., 2012, Hellenthal et al., 2014, Pagani et al., 2016, Barrie et al., 2024, Hu et al., 2023, amongst many more), many of which use these real datasets. Since SparsePainter has been shown with similar (slightly but ignorably lower) accuracy as ChromoPainter for most simulated scenarios, such that we do not think benchmarking with real datasets will provide particular new insight. We agree that doing benchmarking with real datasets is interesting but that could be a separate research project, for example, using these datasets as reference panels for analysing UK Biobank individuals.

A6. The manuscript's first part could benefit from a more clear rewriting, and some figures do not effectively highlight the key points. For example, Figure 1 might be more effectively presented as a concise table. Similarly, Figure 6 could also be condensed in a table. Also, due to the complexity of presenting two similar methods, a simplified visual representation of the methods—especially SparsePainter—could clarify their differences for readers less interested in the technical details. The method section, however, is well-written and easier to follow.

Response: Thank you for your suggestions. We see Fig. 1 as an essential “visual abstract” of the paper, that for readers not familiar with the etymology of Chromosome Painting is a much more concise explanation than the same in text, and the explanatory power would be lost as a table. Fig. 6 is essentially a table and we have reported the

extract SNP positions with shared LDAS and/or AAS signals in Supplementary Table 14, but again the visual nature will help many readers and the Venn diagram would be lost as a table.

We agree on the importance of visually representing the different methods. Fig. 1 was an attempt to explain the conceptual differences between SparsePainter and PBWTpaint, since they differ in intended functionality (i.e. “what are they useful for”). We have added structure to the caption to more clearly highlight this critical function, which we hope addresses your comment.

Changes: We have added structure to the caption of Fig. 1 to more clearly highlight this critical function.

Minor comments:

B1. Figures 2 and 3 are difficult to interpret and would benefit of full re-plotting. The dataset used should be more clearly indicated within the figures, and its naming could be more meaningful.

Response and Changes: Thank you for pointing out the readability of Fig. 2-3. As indicated above, we have replotted these figures to make them easier to read.

B2. The rationale behind selecting specific simulations should be clearer in the main text, with a better description of what each simulation aims to achieve.

Response and Changes: Thank you for suggesting a clearer rationale of each simulation. we have rewritten the Simulation overview (Line187-206) to clearly state both how the simulation differs from others, and the design goals of that.

B3. ChromoPainter has quadratic complexity with the size of the reference. However, Figures 2c and 2d show a slightly more than linear relationship in time and memory usage, but not quadratic. So a phrase such as “fig. 2a-b illustrates that both in theory and practice, ChromoPainter has a quadratic cost as a function of panel size, so scales poorly to larger reference sizes” might need rephrasing.

Response and Changes: Good point - we agree that it is indeed not exactly quadratic in this regime. We have rephrased to “Fig. 2a-b shows the computational scaling of each method. ChromoPainter in theory has a quadratic cost as a function of panel size, and scales poorly to larger reference sizes.” (Line 214-215)

B4. Some wording issues appear in the manuscript:

Page 4, L116: “maga-scale” should likely be “large-scale.”

Response and Changes: We have corrected it as suggested.

Page 4, L123: Two outputs are mentioned, but only one is explained.

Response and Changes: We have added the explanation “The second is the expected fraction of the total genome shared most recently between a target and each reference ancestral individual or population, as used in complex admixture history modelling, e.g. the GLOBETROTTER tool.” (Line 126-129).

Page 5, L155: This introduction also applies to PBWTpaint and might be better placed earlier.

Response and Changes: We agree that this applies to PBWTpaint. This section has been renamed (“From PBWT to accurate sparse local matches”) which does not refer to PBWTpaint or SparsePainter specifically and rewritten to emphasise this (Line 161-163): “As such, shorter matches provide little useful information for tracing local ancestry, but we still need the longest available at a given genomic position.”

Page 6, L175: "Stephen's" should be corrected.

Response and Changes: We have corrected “Stephen’s” as “Stephens”.

B5. The GitHub page for the software is well-organized, and the authors have made the tool accessible and user-friendly.

Response and Changes: Thank you for the kind words! We are always focusing on creating clear documentation and an intuitive interface to make SparsePainter accessible for users of all experience levels.

Reviewer #1 (Remarks to the Author):

The authors have addressed most of our concerns in this revision. However, there are still two comments remaining.

In response to the question “how PBWT was used differently than the way that FLARE did and why it was innovative”, the authors stated that “SparsePainter also uses PBWT technically different from FLARE: SparsePainter uses ‘ReportQLongestMatches’ to ensure the sparsity of matches, which is an improvement to the original PBWT algorithm ‘ReportLongMatches’.” However, based on the description of the “ReportQLongestMatches” algorithm, it seems “ReportQLongestMatches” algorithm is just a few post-processing steps on the matches resulting from the original PBWT algorithm “ReportLongMatches” with different length threshold parameters. Our understanding is that both SparsePainter and FLARE use the same PBWT “ReportLongMatches” algorithm to get matches. Depending on the different length thresholds used, SparsePainter and FLARE get different sets of matches. The authors should talk about why a particular set of matches was needed for SparsePainter and how it would help to improve the local ancestry inference accuracy.

Response and changes: Apologies for the lack of clarity in our wording. We are indeed using “ReportLongMatches” and so in some very reasonable sense are using the same content. However, when we find too few long matches, we reduce the length threshold to systematically search for shorter matches, unlike previous uses. This is essential for **painting** where we require matches at every genome position, in contrast to IBD detection which looks only at genome segments that share a recent relationship.

We have given the reason why we use a different algorithm to get matches (Lines 171-173): “To ensure enough matches are found even in genome regions without long matches, we extend the ‘ReportLongMatches’ algorithm of PBWT with a ‘ReportQLongestMatches’ algorithm which aims to find at least Q longest matches at each position for a target sample i (Methods).”

In the previous revision, we also updated our explanatory paragraph to point out the conceptual difference between chromosome painting and IBD, which explains why we need matches at every genome position and hence the ‘ReportQLongestMatches’ algorithm is designed (Lines 130-132): “Chromosome Painting is conceptually different to identifying haplotypes identical-by-descent (IBD); it assigns every position of the genome to the most recent common ancestor in the reference, without allowing overlaps or conditioning on length and hence the expected age of a sharing event.”

In response to the question of how SparsePainter’s performance is for local ancestry inference on target individuals from the distant admixture events, the authors stated that “We have explored and found that all the LAI methods (including SparsePainter, FLARE, RFMix and ChromoPainter) have significantly decreasing accuracy with more distant

admixture time". However, this statement or any benchmarking results supporting this statement were not found in the manuscript or supplementary material.

Response: Apologies for this oversight. We have done the benchmarking against ChromoPainter and RFMix for target individuals sampled under 13, 25, 50, 75, 100 and 150 generations since admixture. Flare results are not reported because of the same reason as in other benchmarking - that FLARE can only compute IBD for very few SNPs under distant recombination times.

Simulation results show that SparsePainter and ChromoPainter work well under more distant admixture times: Although the Pearson's r^2 reduces moderately by increased admixture times (because match lengths are shorter), the proportion of accurate LAI remains almost unchanged.

Changes: We have added the benchmarking description in Methods section (Lines 686-694): "We also compared the the accuracy of SparsePainter, ChromoPainter and RFMix for target individuals sampled under different generations since admixture.", and reported the results in Supplementary Table 6. We have also interpreted the results in Results section (Lines 263-266): "By showing the local ancestry signal is still preserved to distant admixture times (Supplementary Table 6), e.g. the proportion of accurate LAI retains 93.9% at 150 generations, we can explain why modelling of complex ancient admixture remains robust".

Reviewer #2 (Remarks to the Author):

Response: Thank you again for reviewing our manuscript and supporting your ECRs.

Reviewer #3 (Remarks to the Author):

The authors have responded to all the issues I raised in the initial submission.

Reviewer #3 (Remarks on code availability):

PBWTpaint GitHub repo does not have proper README or instructions on how to use it.

Response and changes: Thank you for your suggestion. We have updated the April 2024 changelog on pbwt highlighting that "The SparsePainter documentation contains full information about how to extract Haplotype Components (HCs) from `pbwt`

-paintSparse` output.”, which provides a link to the documentation on SparsePainter ([https://github.com/YaolingYang/SparsePainter/tree/main/painting-pipeline/Compute%20haplotype%20components%20\(HCs\)](https://github.com/YaolingYang/SparsePainter/tree/main/painting-pipeline/Compute%20haplotype%20components%20(HCs))) that we describe the pipeline to compute HCs from PBWTPaint, and how we use PBWTPaint (with specific focus on command -paintSparse). In addition, the explanation of each parameter for PBWTPaint is simply available through typing ‘pbwt’, which aligns with the design of other functionalities of pbwt.

Separating the documentation is necessary because the pbwt tool performs many functions, of which painting is just one, and the level of detail and examples we provide for painting are not given for the other functions within the repository.

Reviewer #4 (Remarks to the Author):

The paper presents a method whose most noteworthy results are its scalability and ability to handle a large number of reference populations. These strengths are now validated through a wider range of simulations conducted under more challenging, non-ideal conditions.

While restructuring the paper could enhance readability and impact, I leave this decision to the authors and editor.

I have no further comments.

Response: Thank you for reviewing our manuscript and providing us with the valuable suggestions.